# WE CAN HIDE MORE BITS:

## THE UNUSED WATERMARKING CAPACITY IN THEORY AND IN PRACTICE

## ABSTRACT

Despite rapid progress in deep learning–based image watermarking, the capacity of current robust methods remains limited to the scale of only a few hundred bits. Such plateauing progress raises the question: *How far are we from the fundamental limits of image watermarking?* To this end, we present an analysis that establishes upper bounds on the message-carrying capacity of images under PSNR and linear robustness constraints. Our results indicate theoretical capacities are orders of magnitude larger than what current models achieve. Our experiments show this gap between theoretical and empirical performance persists, even in minimal, easily analysable setups. This suggests a fundamental problem. As proof that larger capacities are indeed possible, we train Chunky Seal, a scaled-up version of Video Seal, which increases capacity $4\times$ to $1024$ bits, all while preserving image quality and robustness. These findings demonstrate modern methods have not yet saturated watermarking capacity, and that significant opportunities for architectural innovation and training strategies remain.

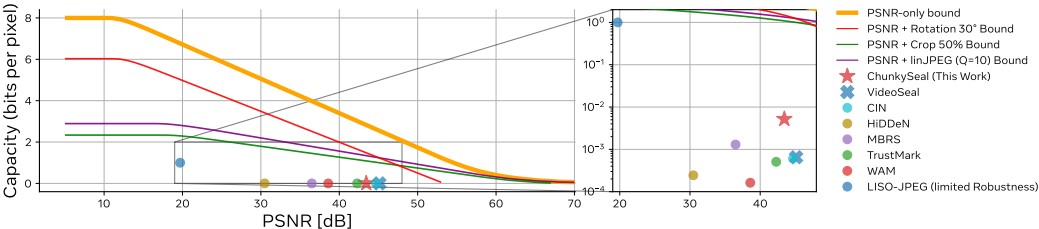

Figure 1: **Existing image watermarking models have capacities well under what this paper suggests to be possible.** Shown are theoretical bounds on watermarking capacity under a PSNR constraint alone (thick line) and in combination with robustness requirements (thin lines). Recent methods operate far below the achievable bounds, often by orders of magnitude, as seen in the log-scale inset. Our proposed **Chunky Seal (1024 bits)** pushes capacity higher than prior work, but is still very far from the theoretical limits, indicating a large potential for future development.

## 1 INTRODUCTION

Invisible image watermarking embeds an *imperceptible* secret message of a *certain capacity* recoverable *under a variety of perturbations*, leading to an inherent capacity-quality-robustness trade-off. Classic methods used hand-crafted tools, such as the mid-frequencies of the discrete cosine transform (Al-Haj, 2007; Navas et al., 2008), discrete wavelet transform (Xia et al., 1998; Barni et al., 2001) or a combination of them (Navas et al., 2008; Feng et al., 2010; Zear et al., 2018). Deep learning led to significant improvements in all three dimensions via attacking fixed decoders (Vukotić et al., 2018; Fernandez et al., 2022), or via end-to-end training of the embedder and decoder (Mun et al., 2017; Zhu et al., 2018; Tancik et al., 2020; Bui et al., 2023a; Xu et al., 2025; Sander et al., 2025). Yet, despite these techniques, it seems that progress has stagnated. State-of-the-art methods successfully embed around $100-200$ bits in a relatively imperceptible way (i.e., Peak Signal-to-Noise Ratio, PSNR, above $40\,\text{dB}$) while robust to perturbations. Improvements in quality and robustness continue, but they are only marginal, leading many to believe we are nearing the limits of what is possible.

Image watermarking indeed may already be a solved problem. Unlike generative or discriminative models that can improve as data and parameters are scaled, watermarking has an inherent performance ceiling. Given an image resolution and a set of robustness constraints, there is a finite amount of information that can be embedded imperceptibly. The existence of this limit and the converging empirical performance of recent models naturally leads to a critical question: **Have we already reached the theoretical ceiling of watermarking performance?**

To answer this question, we need to know what this limit actually is and to measure how close our models are to it. We address these challenges in the current paper and offer the following findings:

i. We propose bounds on the capacity of watermarking under a PSNR constraint and robustness to linear augmentations, indicating capacities orders of magnitude larger than seen in practice.

ii. Watermarking models are trained with constraints we cannot directly analyse so we retrain Video Seal (Fernandez et al., 2024), a SOTA image and video watermarking model, to match our simplest theoretical setup: watermarking a single gray image under only a PSNR constraint. Yet, Video Seal fails to encode even $1024$ bits, when we successfully encode $2048$ bits with a linear model, $32{,}768$ bits by tiling lower-resolution watermarks, and $456{,}509$ bits with a handcrafted model. This indicates severe structural limitations.

iii. With the standard quality and robustness constraints, we train Chunky Seal, a simple scale-up of Video Seal, which embeds $1024$ bits while maintaining similar robustness and image quality.[1]

Therefore, our theory and experiments show that **it is possible to achieve much higher capacities than we currently have, although that might require innovation in architectures and training.**

## 2 BOUNDS ON WATERMARKING CAPACITY

We first discuss previous approaches to watermarking capacity in Section 2.1. We then model images as points on a high-dimensional grid, where capacity is determined by the number of unique points that satisfy imperceptibility and robustness constraints. Using this model, we first establish the absolute maximum information capacity (Section 2.2), then apply a PSNR constraint (Sections 2.3 and 2.4), and subsequently incorporate robustness to transformations like cropping, rescaling, and rotation (Section 2.5). We conclude by exploring the impact of data distribution on capacity (Section 2.6).

### 2.1 RELATED WORK ON THEORETICAL MODELS OF WATERMARKING CAPACITY

Previous work on watermarking capacity largely relied on unrealistic assumptions like Gaussian noise (Costa, 1983; Cohen and Lapidoth, 2002; Chen and Wornell, 2002). More practical approaches were limited to small-magnitude perturbations (Moulin and O'Sullivan, 2003; Moulin and Koetter, 2005; Somekh-Baruch and Merhav, 2004) or specific geometric transformations (Merhav, 2005). Rather than these information-theoretic methods, which view the problem as power-limited communication over a super-channel with a state that is known to the encoder, our work takes a geometric approach allowing us to study more realistic conditions. Extended related works on image watermarking methods and theoretical approaches are discussed in App. A.

### 2.2 ABSOLUTE CAPACITY OF THE IMAGE SPACE

Watermarking embeds a message $m$ into an image $\boldsymbol{x}$. Since each message must correspond to a distinct encoded image, the number of unique messages, that is, the watermarking *capacity* is limited by the number of distinct images. An $l$-bit message requires at least $2^l$ such images. We represent an image as a vector of length $cwh$, where $c$ is the number of channels, $w$ is the width and $h$ the height, with each element having $2^k$ discrete levels when using $k$-bit colour depth. The tuple $(c, w, h, k)$ defines an *image format*. The set of all possible images in this format is $\mathcal{I} = \{0, 1, \ldots, \rho\}^{cwh}$ with $\rho = 2^k - 1$, which can be thought of as a finite grid[2] of points in $\mathbb{R}^{cwh}$. This immediately gives us a trivial upper bound on watermarking capacity: since each message must correspond to a distinct watermarked image, it is not possible to embed more messages than there are distinct images.

> **Bound 1: Absolute capacity of the image.** The capacity of images in the format $(c, w, h, k)$ is
>
> $$\texttt{capacity[in bits]} = \log_2 |\mathcal{I}| = \log_2 \left( (2^k)^{cwh} \right) = cwhk \ \text{ bits.}$$

---

[1]Chunky Seal code and checkpoints will be released.

[2]The set of valid images is a finite subset of a lattice in $\mathbb{R}^{cwh}$, i.e., $\mathcal{I} = [0, \rho]^{cwh} \cap \{\sum_{i=1}^{cwh} a_i \boldsymbol{e}_i \mid a_i \in \mathbb{Z}\}$ with $\boldsymbol{e}_i$ the $i$-th unit vector, but we use the term *grid* for readability since no deeper lattice theory is required.

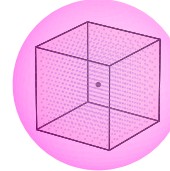 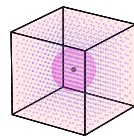 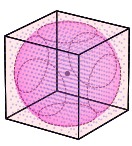 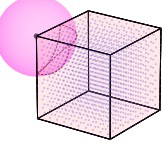

(a) Cube fully inside sphere (low PSNR, Bound 2)  (b) Sphere fully inside cube (high PSNR, Bounds 3 and 4)  (c) Non-trivial intersection (medium PSNR, Bounds 6 and 5)  (d) Sphere at corner, i.e., arbitrary image (Section 2.4)

Figure 2: **The box-ball configurations of the PSNR-only constraint.** The cube $C_{\mathcal{I}}$ is the set of all images and the sphere is the PSNR ball centred at the cover. Their intersection determines the set of feasible watermarked images, with the cardinality being the watermarking capacity. **(a)**, **(b)** and **(c)** are the cases with the cover image $x$ at the centre of the cube $C_{\mathcal{I}}$ (gray image, resulting in highest capacity, Section 2.3). **(d)** is the case of the worst-case cover $x$, i.e., at the corner of $C_{\mathcal{I}}$ (Section 2.4).

Bound 1 simply states that the maximum number of embeddable bits is the uncompressed size of the image in bits. We next introduce imperceptibility and robustness to measure their effect on capacity.

## 2.3 Capacity under a PSNR constraint

A standard way to quantify distortion is the *peak signal-to-noise ratio (PSNR)*, measured in dB. Requiring a minimum PSNR $\tau$ between the cover $x$ and the watermarked image $\tilde{x}$ is equivalent to bounding their $\ell_2$ distance (see App. B for the full derivation):

$$\mathrm{PSNR}(x, \tilde{x}) \geq \tau \iff \|x - \tilde{x}\|_2 \leq \epsilon(\tau), \text{ with } \epsilon(\tau) = \rho\sqrt{cwh}\,10^{-\tau/20}. \tag{1}$$

Interpreting PSNR as an $\ell_2$-ball constraint gives us an avenue for measuring the message-carrying capacity under it by considering the amount of integer points inside both the cube and this ball. Counting how many such points exist is not trivial, and we analyse the three possible cases (see Figure 2): *i.* the ball is so large that it contains the entire cube (very low $\tau$); *ii.* the ball is small enough to lie fully inside the cube (high $\tau$); *iii.* the ball and cube partially overlap (medium $\tau$). We begin by assuming the cover image $x$ lies at the centre of the admissible range, i.e., $x_g = 2^{k-1}\,\mathbf{1}$ as then the volume of the intersection (and thus the capacity) is maximized. In Section 2.4 we will extend the analysis to arbitrary images.

### 2.3.1 Cube in ball (low PSNR)

When $\tau$ is low, $\epsilon(\tau)$ is large and the ball contains the entire cube $C_{\mathcal{I}} = [0, \rho]^{cwh}$. The PSNR constraint does not rule out any images, so the capacity is just the absolute maximum (Bound 1):

**Bound 2: Gray image, PSNR constraint (low PSNR).** The capacity of a gray image $x_g$ under a very low minimum PSNR threshold $\tau$ is capacity[in bits] $= cwhk$.
**Bound validity:** When $\epsilon(\tau) \geq \rho/2\sqrt{cwh}$, or equivalently when $\tau \leq 20\log_{10} 2 \approx 6.02$ dB.

### 2.3.2 Ball in cube (high PSNR)

When $\tau$ is high, the PSNR ball is fully inside the cube. The capacity is the number of integer points inside the ball, a problem with no general closed form. In high dimensions $cwh$ and sufficiently large radii $\epsilon(\tau)$, this is well approximated by the ball volume $\mathrm{Vol}\,B_{cwh}[\cdot, \epsilon(\tau)]$ (see Appendix C).

**Bound 3: Gray image, PSNR constraint (high PSNR, volume approximation).** The capacity of a gray image $x_g$ under a high minimum PSNR threshold $\tau$ is approximately $\log_2 \mathrm{Vol}\,B_{cwh}[\cdot, \epsilon(\tau)]$.
**Bound validity:** When the ball is fully inside the cube, i.e. $\epsilon(\tau) \leq \rho/2$ (i.e., $\tau \geq 20\log_{10}(2\sqrt{cwh})$) and $\epsilon(\tau)$ large enough for accurate volume approximation (see Bound 8 for small $\epsilon$).

For small radii the volume approximation becomes inaccurate. However, then there are relatively few integer points in the ball and we can explicitly count them, as long as the dimension $cwh$ of the ambient space is not too high. Instead of brute-force enumeration (which scales poorly), we use a method introduced by Mitchell (1966) leveraging symmetries for efficient counting (see Algorithm 2).

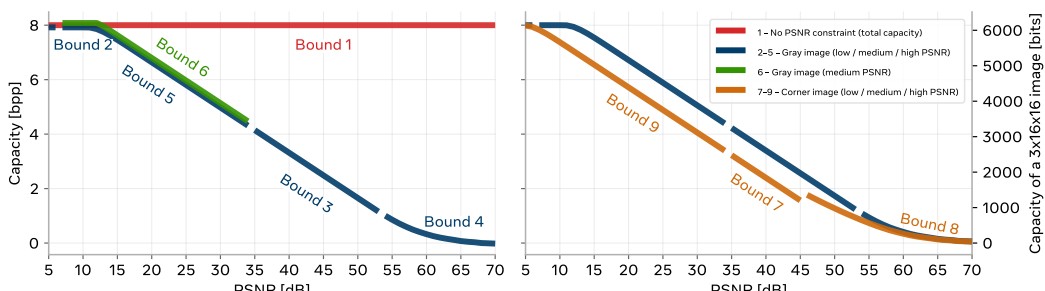

Figure 3: **Watermarking capacity under PSNR constraints for a 16×16px 3-channel cover image without robustness requirements.** We show the case where the cover lies at the centre of the pixel range (left, Sections 2.2 and 2.3) and at the extreme corner where pixels are saturated (right, Section 2.4). In both cases we plot the family of bounds we established ranging from the trivial total capacity (Bound 1) to volume approximations (Bounds 3 and 7), exact lattice counts (Bounds 4 and 8), and numerical integration for partial overlap (Bounds 5 and 9). In the right panel, the arbitrary cover image case lies below the centred gray image case by at most $cwh$ bits i.e., at most one bit per pixel (1 bpp). The discontinuity between Bounds 7 and 8 is due to the volume approximation undercounting the integer points on the faces of the cube.

**Bound 4: Gray image, PSNR constraint (high PSNR, exact count).** For small $\epsilon(\tau)$ the capacity is

$$\texttt{capacity[in bits]} \;=\; \log_2 \texttt{PointsInHypersphereMitchell}(\texttt{dim} = cwh, \texttt{radius} = \epsilon(\tau)).$$

**Bound validity:** When $\epsilon(\tau) \leq \rho/2$ (i.e., $\tau \geq 20\log_{10}(2\sqrt{cwh})$) and $\epsilon(\tau)$ small enough that exact counting is computationally feasible.

We use Bound 4 whenever we can evaluate Algorithm 2 in reasonable time, and otherwise Bound 3. As shown in Figure 7, the transition between the two regimes is smooth.

### 2.3.3 NON-TRIVIAL INTERSECTION (MEDIUM PSNR)

For intermediate PSNR values $\tau$, $B_{cwh}\left[\boldsymbol{x}_g, \epsilon(\tau)\right]$ and $C_{\mathcal{I}}$ intersect non-trivially. We can approximate this count by the volume of the intersection, using the same volume-based method as in Bound 3. One can use exact volume computation (see Bound 5 in Appendix E), though this tends to be numerically unstable. In practice, a simpler upper bound approximates it well:

**Bound 6: Gray image, PSNR constraint (medium PSNR, approximation).** The capacity of a gray image under minimum PSNR $\tau$ is upper-bounded by $\min[$Bound 2, Bound 3$]$.
**Bound validity:** When $\rho/2 \leq \epsilon(\tau) \leq \rho/2\sqrt{cwh}$, or equivalently $20\log_{10} 2 \leq \tau \leq 20\log_{10}(2\sqrt{cwh})$.

As shown in Fig. 3 left, this simple upper Bound 6 closely tracks the exact Bound 5. Thus Bound 6 is the practical choice going forward, while Bound 5 is provided in the appendix for completeness. Figure 3 left illustrates all the bounds from this section for a $16\times16$px image. At $45$ dB these bounds give us roughly $2000$ bits of capacity (more than 2.5 bpp): orders of magnitude more than the $0.001$ bpp we see in practice (Figure 1).

### 2.4 FROM CENTRAL GRAY IMAGE TO ARBITRARY COVER IMAGES

In Section 2.3 we assumed the cover lies at the centre of the pixel range, thereby maximizing the volume of the intersection between the PSNR ball and the cube $C_{\mathcal{I}}$. Real images, however, may be anywhere in $C_{\mathcal{I}}$. Being at the corner of $C_{\mathcal{I}}$ minimizes overlap with the ball and thus provides a lower bound valid for any image. When $\epsilon$ is not too large, exactly $1/2^{cwh}$ of the PSNR ball centred at a corner of $C_{\mathcal{I}}$ remains inside $C_{\mathcal{I}}$. Although this may seem drastic, the penalty is in fact modest: at most $cwh$ bits, i.e., one bit per pixel. In Appendix F we provide the formal bounds for this corner setting. Bound 7 adapts Bound 3, the volume approximation when the ball is fully in the cube. Bound 8 is the analogue of Bound 4, i.e., exact counting for small $\epsilon(\tau)$. Bound 9 parallels Bound 5 for the case when numerical integration is needed. As shown in Figure 3, the gap from the gray-only image bounds is at most $1$ bpp, thus: **Watermarking with a PSNR constraint should allow for capacity upwards of 2 bpp and does not explain the low capacities we observe in practice.**

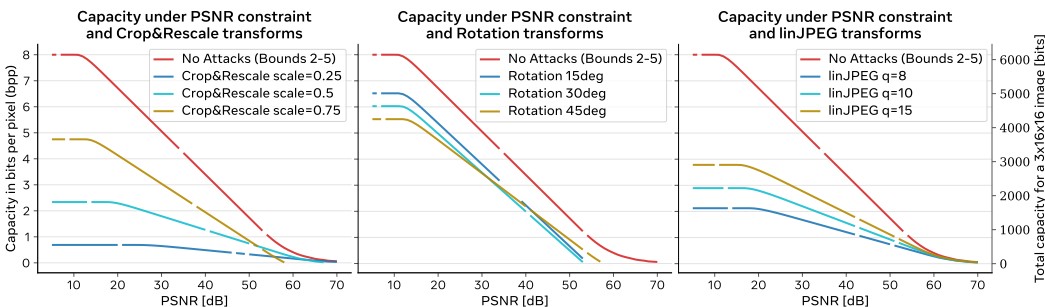

Figure 4: **Impact of robustness constraints on watermarking capacity.** Capacity (in bits per pixel) under a PSNR constraint is shown for three families of transformations: *Crop&Rescale* (left), *Rotation* (center), and *Linearized JPEG* (right), using the heuristic bounds (Bounds 10 to 12). The red lines show the PSNR-only capacity bounds without robustness constraints Bounds 2 to 4 and 5. Each transformation reduces capacity in proportion to its severity: smaller crop scales, larger rotations, or lower JPEG quality factors. Across all cases, robustness constraints reduce but do not eliminate the large theoretical capacity gap with current watermarking methods.

### 2.5 Adding robustness constraints

In practice, watermarking must balance imperceptibility with robustness: the message should survive common processing, like compression, resizing, cropping, rotation, etc. In our model, we consider linear transformations, which encompass most transformations used in practice. We also develop LinJPEG, a linearized version of JPEG, allowing us to study the effects of compression in the same setting (see Appendix G.4 for the construction). Take a linear transformation $M \in \mathbb{R}^{cwh \times cwh}$ that maps an image $\boldsymbol{x}$ to a transformed $M\boldsymbol{x}$ and a quantization operation $\mathbb{Q}$ (element-wise rounding or floor operation) to map the pixel values of $M\boldsymbol{x}$ to the valid images $\mathcal{I}$. Hence, we have the final transformed image $\boldsymbol{x}' = \mathbb{Q}[M\boldsymbol{x}]$. We need to find the subset of the possible watermarked images under only the PSNR constraint that map to unique valid images after applying $M$ and $\mathbb{Q}$ to them. The main complication in this setup is that $\mathbb{Q}$ is non-linear.

**Heuristic bounds.** A simple approach is to take a volumetric approach akin to Bounds 3, 5, 7 and 9. We factor in how $M$ changes the volume and account for directions compressed by the transformation which destroy capacity as different watermarked images get collapsed together. We also account for directions fully collapsed by $M$ when it is singular. Finally, the stretched directions might result in some watermarked images being outside $C_{\mathcal{I}}$ after the transformation, leading to them being clipped. Bounds 10 to 12 use a heuristic based on the singular values of $M$ to account for the effect on capacity. Refer to Appendix G.2 for details. In Figure 4 we plot these bounds for robustness to rotation, cropping followed by rescaling and LinJPEG, showing that even under the most aggressive cropping, we should expect around 0.5 bpp or almost 100,000 bits for 256×256px images.

**Conservative bounds.** We can show cases where these heuristic bounds under-approximate and cases where they over-approximate the true capacity, e.g., Figures 8 and 9. Thus, the true capacity under linear transformation could be much lower than these bounds predict. To ensure that this is not the case, we develop an actual lower bound: Bound 13. While we reserve the details for Appendix G.3, this bound is based on over-approximating the set of images that can be quantized by $\mathbb{Q}$ to the same image after $M$ is applied to them. As a result, Bound 13 is extremely conservative and unrealistic. We believe that despite Bounds 10 to 12 not being valid lower bounds, they are much closer to the true capacity. Still, we report the conservative bound in Table 2: the most aggressive crop still leaves at least 904 bits for 256×256px images. For the other augmentations, the conservative capacity is much higher. Therefore, **robustness to geometric transformations and compression significantly reduces the capacitybut cannot fully explain the low watermarking capacity of current models.**

### 2.6 From single cover images to datasets and data distributions

In a blind watermarking setup, the decoder must operate without access to the original cover image, creating potential collisions: if multiple natural images (i.e., potential covers) are very close to each other in pixel space, a watermarked version of one cover could be identical to a watermarked version of another. To prevent such ambiguity, the total set of watermarked images within a given region

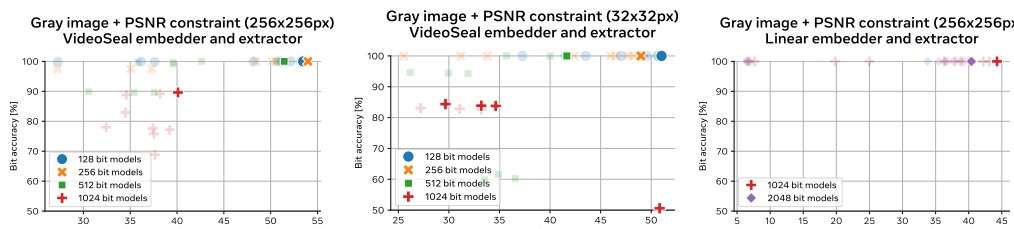

Figure 5: **Video Seal fails to learn how to embed** $1024$ **bits into a gray image with only PSNR constraint, whereas a linear embedder and extractor learn how to embed** $2048$ **bits.** *(Left.)* Video Seal trained on a single solid gray image with only the detector and MSE losses. It learns to embed up to $512$ bits but fails to embed $1024$ bits. *(Centre.)* Same setup as the left plot but trained on a reduced $32{\times}32$px resolution. The performance is similar; hence, the Video Seal architecture fails to make use of the full resolution. *(Right.)* Replacing the embedder and decoder of Video Seal with a single linear layer each achieves 100% bit accuracy and PSNR above $40$ dB for both 1024 and 2048 bits: demonstrating that Video Seal indeed has structural limitations. The results for the sweeps over the learning rate and $\lambda_i$ are shown, with the best models highlighted.

(like the PSNR ball) must be partitioned among all the potential covers it contains. If there are $N$ possible covers, the capacity for each is reduced by $\log_2(N)$ bits. We estimate $N$ using neural compression models like VQ-VAE (Van Den Oord et al., 2017) and VQGAN (Esser et al., 2021), which upper-bound the number of perceptually distinct images. For instance, a $256{\times}256$px image can be compressed into a $32{\times}32$ latent with a 1024-entry codebook (Muckley et al., 2023). This representation can express at most $1024^{32\times32} = 2^{10240}$ distinct images. Conservatively assuming all could fall in the PSNR ball of the considered image, capacity is reduced by $10,240$ bits, or about $0.05$ bpp, on top of the 1 bpp loss from Section 2.4. Thus, from this perspective, **the data distribution has only a negligible effect on watermarking capacity and cannot explain the low performance of current models.** This aligns with prior findings for Gaussian channels that decoder knowledge of the cover does not affect capacity (Costa, 1983; Chen and Wornell, 2002; Moulin and O'Sullivan, 2003).

## 3 Empirical performance is much lower than predicted

Section 2 showed that capacities of over 2 bpp at PSNR of $40$ dB without robustness constraints, and of 0.5 bpp with robustness, are possible. Even under the very conservative Bound 13 we still would expect capacities of at least 0.01 bpp. However, in practice, the models reported in the literature have significantly lower capacities (less than 0.001 bpp, Figure 1). To understand the cause of this gap, this section asks: **are existing models significantly under-performing relative to what is possible in practice, or are our bounds too unrealistic?** There are five possible explanations of the large discrepancy between the performance we see in practice (Figure 1) and the bounds in Section 2:

- *A.* **Real models might be near-optimal if we consider advanced robustness constraints.**
- *B.* **Real models might be near-optimal if we consider advanced perceptual constraints.**
- *C.* **Real models might be near-optimal if we consider real-world image distributions.**
- *D.* **Our bounds overestimate capacity and cannot be approached empirically.**
- *E.* **We can do much better and push the Pareto front well beyond the current state-of-the-art.**

To understand the cause of the gap between theoretical and real-world performance in image watermarking, we need to find out which of these hypotheses is the underlying cause. If it is *A.*, *B.*, *C.*, *D.*, or a combination of them, then it is possible that, indeed, the best current models are close to what is ultimately possible and we can expect only marginal further improvements. On the other hand, if the cause is *E.*, then that means that there is plenty of space for significant improvements.

### 3.1 The real-world complexity does not explain the performance gap

Let's first address cases *A.*, *B.*, *C.*, i.e., that our bounds cannot capture the complexity of the robustness, quality and data constraint with which real models are trained. While we cannot bring the real-world complexity to our analytical bounds, we can bring the models to the simplified theoretical setup.

Table 1: **Video Seal fails to learn how to embed 1024 bits into a gray image with only PSNR constraint while a linear embedder and extractor learn how to embed 2048 bits.** Numerical results for the best-performing runs from Figure 5 and their respective hyperparameters, as well as the handcrafted embedder/decoder from Equation (2) at four representative PSNR values.

| | Message size | Message size if tiled to 256x256px | PSNR | Bit acc. | $\lambda_i$ | lr |
|---|---|---|---|---|---|---|
| VideoSeal (256x256px, 600 epochs) | 128 bits | | 53.45 dB | 100.00% | 0.5 | 5e-4 |
| | 256 bits | | 53.98 dB | 100.00% | 1.0 | 5e-4 |
| | 512 bits | | 51.45 dB | 100.00% | 0.5 | 5e-4 |
| | 1024 bits | | 40.10 dB | 89.63% | 1.0 | 5e-5 |
| VideoSeal (32x32px, 600 epochs) | 128 bits | 8192 bits | 51.02 dB | 100.00% | 1.0 | 5e-4 |
| | 256 bits | 16384 bits | 48.98 dB | 100.00% | 1.0 | 5e-4 |
| | 512 bits | 32768 bits | 41.66 dB | 100.00% | 1.0 | 5e-5 |
| | 1024 bits | 65536 bits | 29.66 dB | 84.39% | 0.1 | 5e-5 |
| | 1024 bits | 65536 bits | 33.20 dB | 83.86% | 0.5 | 5e-5 |
| | 1024 bits | 65536 bits | 34.63 dB | 83.78% | 1.0 | 5e-5 |
| | 1024 bits | 65536 bits | 50.83 dB | 50.60% | 0.5 | 5e-4 |
| Linear (256x256px, 50 epochs) | 1024 bits | | 44.28 dB | 100.00% | 20.0 | 5e-4 |
| | 2048 bits | | 40.40 dB | 100.00% | 12.0 | 5e-4 |
| Handcrafted | 623232 bits | | 36.00 dB | 100.00% | | |
| | 551948 bits | | 38.00 dB | 100.00% | | |
| | 456509 bits | | 42.00 dB | 100.00% | | |
| | 311616 bits | | 48.00 dB | 100.00% | | |

More concretely, we take the simplest of setups: a single gray image with a PSNR constraint, as in Section 2.3. We will use Video Seal as the base for our experiments (Fernandez et al., 2024), originally introduced as an image watermarking model with frame copying that generalizes to video. It was first demonstrated with a 96-bit capacity and was recently extended to a 256-bit open-source version, which we use as the strongest available baseline. To match the setup of Section 2.3, we replace the dataset with a single solid gray image, remove all perceptual constraints but the MSE loss and remove all augmentations. We first retrain it for $n_{\text{bits}} = 128, 256, 512,$ and $1024$ bits. We have hereby reduced the task to simply find a way to encode $n_{\text{bits}}$ into a single fixed image. From Figure 3 we expect capacities of around $600,000$ bits at $40\,$dB in this setup. Thus, the model should easily learn these much lower $n_{\text{bits}}$. We train with AdamW (Loshchilov and Hutter, 2019) with batch size 256 for 600 epochs, 1000 batches per epoch, cosine learning rate schedule with a 20-epoch warm-up, similarly to Video Seal. We sweep over the learning rate (5e-4, 5e-5, 5e-6) and $\lambda_i$, the MSE loss weight (0.1, 0.5, 1.0), with LR=5e-5 and $\lambda_i = 0.5$ being the values used for training Video Seal.

The results of training Video Seal on a single gray image can be seen in Figure 5 left and Table 1. There are runs for the 128, 256 and 512 bit models that do achieve 100% bit accuracy and PSNR values above 42dB. However, Video Seal cannot even get to $1024$ bits, far from what we expect from the bounds. This is surprising: the model cannot approach the theoretical bounds even after removing the complexities that supposedly make watermarking difficult. This means that neither ***A.***, ***B.*** nor ***C.*** can explain why we see such a gap between the theoretical and real-world performance.

### 3.2 OUR SIMPLEST BOUNDS ARE ACHIEVABLE, YET MODELS STRUGGLE TO GET NEAR THEM

Section 3.1 showed that Video Seal cannot match the capacity predicted by the bounds in Section 2.3 even when trained only on a single gray image and with no augmentations. Thus, the complexity of real world watermarking cannot explain the gap between the theoretical and real-world performance. This leaves us with two options: ***D.*** our bounds are wrong and unachievable, or ***E.*** our models are under-performing. There are a couple simple experiments that can demonstrate that we can get much closer to the bounds in Section 2.3 and hence ***D.*** also does not explain the gap.

**Linear embedder and extractor.** We trained a simple linear embedder and extractor. The embedder gets the 1024 bit message (shifted and scaled to $-1$ and $+1$ values) and produces a $256{\times}256{\times}3$ watermark residual which gets added to the original gray image. Similarly, the decoder is a linear layer from the flattened $256{\times}256{\times}3$ image to 1024 outputs, which are thresholded to recover the message. We train only for 50 epochs, with the same learning rate values and $\lambda_i \in \{4, 8, 12, 20\}$.

Table 2: **Conservative capacity bounds under robustness constraints for PSNR 42 dB.** These values are calculated via Bound 13 and are strongly conservative lower bounds on the capacity that is achievable while maintaining robustness to the respective transformations and PSNR under $42\,\text{dB}$.

| | | Conservative capacity | |
| Augmentation | bpp | for $16{\times}16$px | for $256{\times}256$px |
| --- | --- | --- | --- |
| Horizontal Flip | 3.064 | 2,352 bits | 602,353 bits |
| Crop&Rescale 50% | 0.015 | 11 bits | 3,013 bits |
| Crop&Rescale 75% | 0.005 | 3 bits | 904 bits |
| LinJPEG q=10 | 0.136 | 104 bits | 26,757 bits |
| LinJPEG q=15 | 0.137 | 105 bits | 27,020 bits |
| Rotation 30deg | 0.075 | 57 bits | 14,676 bits |
| Rotation 45deg | 0.083 | 64 bits | 16,401 bits |

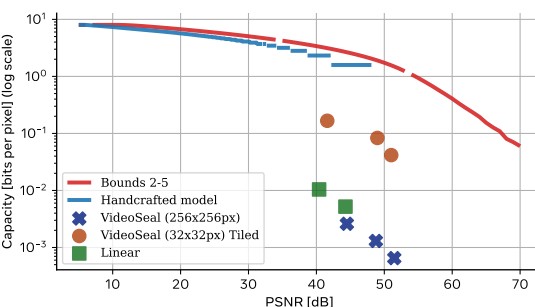

Figure 6: **Simple models outperform Video Seal on a gray image with only a PSNR constraint.** Experiments from Section 3 compare our theoretical bounds (Section 2.3) against trained models. Video Seal falls well below the predictions, while a linear model performs slightly better and a tiled $32{\times}32$px Video Seal is even better. Our handcrafted model nearly matches the bound.

The results in Figure 5 right and Table 1 show the linear layer learns what Video Seal could not: 100% bit accuracy for $1024$ bits with PSNR of $44\,\text{dB}$. We also trained a linear model for $2048$ bits which achieved 100% bit accuracy. This shows that capacities beyond $512$ bits are possible in practice (at least for a gray image and no robustness) and are learnable via gradient descent. All one needs is the right architecture.

**Lower-resolution training and tiling.** Our experiments reveal that Video Seal does not exploit the additional degrees of freedom available at higher image resolutions. When trained at $256{\times}256$px, the model achieves essentially the same capacity and PSNR as when trained at $32{\times}32$px (see Section 3.1 and Table 1). To verify this, we train Video Seal in the setup of Section 3.1 at $32{\times}32$px using the same learning-rate and $\lambda_i$ sweeps for 600 epochs. As shown in Figure 5 (centre) and Table 1, the performance at $32{\times}32$px is nearly identical to that at $256{\times}256$px: the 512-bit model reaches 100% bit accuracy with $41.7\,\text{dB}$, despite operating on $64\times$ fewer pixels. In other words, the effective capacity we observe at $256{\times}256$px is comparable to what one would expect around $20{\times}20$px, confirming that the architecture fails to utilize the available resolution.

Because this setup does not require robustness to geometric or valuemetric transformations and we consider only gray images, we can use the $32{\times}32$px model to demonstrate that higher capacities are possible. A simple tiling strategy suffices: each tile is embedded with an independent secret using the same model. The decoder similarly is applied per patch with the individual decoded messages concatenated to obtain the final combined message. Using $256{\times}256$px as the reference size, tiling yields $64\times$ the capacity of the base $32{\times}32$px model. Thus, tiling the 512-bit model—which already achieves 100% accuracy at $41.7\,\text{dB}$—produces a watermark with $32,768$ bits total capacity while maintaining the same PSNR (which is resolution-independent). This effective capacity of $32,768$ bits is already much closer to our bound of roughly $600,000$ bits, though still only about $0.167$ bpp.

It is interesting that the model could not learn at the $256{\times}256$px resolution even for $1024$ bits when it is clear that it is possible to embed $32,768$ bits as seen here. More importantly, this shows that our bounds are not that far off and capacities of at least $32,768$ bits are indeed possible.

**Handcrafted embedder and extractor.** We can do even better by manually crafting an embedder and extractor. The key observation is that mapping a hypercube to binary messages is easy. Take the ball of radius $\epsilon(\tau) = \rho\sqrt{cwh}\,10^{-\tau/20}$ from Equation (1). The half-side of the largest cube that can fit in this ball is $d = \epsilon(\tau)/\sqrt{cwh} = \rho\,10^{-\tau/20}$. We have that each edge of the box contains a $cwh$-dimensional grid of $q = 2\lfloor d \rfloor + 1 = 2\lfloor 2^k\,10^{-\tau/20} \rfloor + 1$ points per side. Hence, that gives us

|  | Chunky Seal (ours) | Video Seal 256bits |
|---|---|---|
| Capacity | 1024 bits
0.0052 bpp | 256 bits
0.0013 bpp |
| Embedder size | 1022.7M | 11.0M |
| Extractor size | 773.7M | 33.0M |
| PSNR ↑ | 45.32±2.16 | 44.42±2.21 |
| SSIM ↑ | 0.995±0.006 | 0.996±0.003 |
| MS-SSIM ↑ | 0.997±0.002 | 0.997±0.001 |
| LPIPS ↓ | 0.0085±0.0067 | 0.0019±0.0011 |
| Bit acc. Identity | 99.74±0.28% | 99.90±0.21% |
| Bit acc. Flip | 99.65±0.34% | 99.89±0.24% |
| Bit acc. Rotate (≤10°) | 98.27±2.10% | 98.84±1.10% |
| Bit acc. Resize (71–95%) | 99.74±0.28% | 99.90±0.21% |
| Bit acc. Crop (77–95%) | 98.25±1.75% | 98.04±1.57% |
| Bit acc. Brightness (0.5–1.5×) | 98.99±1.87% | 98.67±2.67% |
| Bit acc. Contrast (0.5–1.5×) | 99.54±0.51% | 99.56±0.45% |
| Bit acc. JPEG (Q 50–80) | 98.79±0.75% | 99.74±0.47% |
| Bit acc. Gaussian Blur (k≤9) | 99.74±0.28% | 99.90±0.22% |
| Bit acc. Overall | 99.15±0.63% | 99.31±0.60% |

Table 3: **Chunky Seal performance on images from SA-1B (Kirillov et al., 2023) at their original resolution.** Chunky Seal has much higher capacity (1024 bits) than Video Seal while preserving its image quality and robustness on a wide variety of transformations. The improvement is driven by scaling the model size and its training. Extended results on SA-1B (Kirillov et al., 2023) and COCO (Lin et al., 2014) as well as qualitative results, are reported in Appendix J.1

total capacity in bits of

$$\log_2\left[\left(2\left\lfloor 2^k \ 10^{-\tau/20}\right\rfloor + 1\right)^{cwh}\right] = cwh \log_2\left[2\left\lfloor 2^k \ 10^{-\tau/20}\right\rfloor + 1\right] = cwh \log_2 q, \quad (2)$$

or $\log_2 q$ bits per pixel. See Figure 6 for a plot of that for different PSNR values. For $42\,\mathrm{dB}$, and images of $256{\times}256\mathrm{px}$ that gives us a capacity of $456{,}509$ bits (see Table 1) almost $14\times$ what we could embed with the $32{\times}32\mathrm{px}$ tiling approach. Moreover, it gets us close to the theoretical bound.

Therefore, we can get much closer to the boundary, at least in the solid gray image case with PSNR constraint and no robustness requirements. Thus, case **D.**, that our bounds are wrong and impossible to achieve, is unlikely. This leaves us with one possible explanation as to why models in practice do not exhibit performance anywhere near what our theory predicts. That would be option **E: Our models are likely significantly underperforming relative to what is possible in practice. We likely can do much better and push the Pareto front well beyond the current state-of-the-art.**

## 4  BETTER PERFORMANCE IN PRACTICE IS POSSIBLE: CHUNKY SEAL

While it remains possible that current models approach a theoretical limit under robustness and quality constraints, training a watermarking model with comparable quality and robustness but with substantially higher capacity would decisively rule this out. We take Video Seal (Fernandez et al., 2024) as the base model and train it for 1024 bits. We increased the embedding dimension to 2048, the U-Net channel multipliers from $[1, 2, 4, 8]$ to $[4, 8, 16, 32]$, and enabled watermarking in all three channels, not just the luma (Y) channel. This results in an embedder $90\times$ larger than the original Video Seal embedder. The ConvNeXt (Liu et al., 2022) extractor was similarly scaled: we increased the depths for each stage from $[3, 3, 9, 3]$ (as in ConvNeXt-tiny) to $[3, 3, 27, 3]$ (as in ConvNeXt-base), with their dimensions increased from $[96, 192, 384, 768]$ to $[256, 512, 1024, 2048]$. The stride of the first layer was reduced from 4 to 2. This results in an extractor that is $23\times$ larger than the original Video Seal extractor. Due to its significantly increased size, we name this model Chunky Seal. We train it at the original $256{\times}256\mathrm{px}$ resolution. We apply gradient clipping with a maximum norm of 0.01, which proved critical for stabilizing training.

As shown in Table 3, Chunky Seal shows image quality and robustness comparable to Video Seal across a wide range of distortions, while providing a $4\times$ **higher message capacity** (1024 vs. 256 bits). Despite its much larger capacity, Chunky Seal maintains nearly identical image quality across all metrics, and only slightly higher LPIPS. The robustness results further confirm that Chunky Seal sustains high bit-accuracy across transformations such as rotation, resizing, cropping, brightness and contrast changes, JPEG compression, and blurring, closely matching Video Seal. We emphasize that these results were achieved *without hyperparameter tuning*, whereas Video Seal was extensively optimized for quality and robustness. **Achieving $4\times$ the capacity per pixel with comparable robustness and quality through simple scaling strongly suggests that substantially higher capacities are within reach using improved architectures and training strategies.**

## 5 DISCUSSION AND CONCLUSIONS

Higher watermarking capacities open up new avenues for content provenance. Instead of using a watermark to retrieve a C2PA manifest from a third-party database (Collomosse and Parsons, 2024), we could embed the entire manifest, eliminating the need for a registry. Beyond this, improvements in capacity can be traded for greater robustness or higher image quality, depending on the application. The fact that our theoretical capacity bounds are an order of magnitude higher than even the best existing models also helps explain why applying and detecting multiple watermarks on a single image is feasible, as demonstrated by (Petrov et al., 2025).

Despite achieving substantially higher capacities than prior models, Chunky Seal still remains far from the theoretical bounds established in this work. Our controlled experiments show that this gap cannot be attributed to factors such as data distribution, resolution, or augmentations. Instead, the evidence consistently points to limitations in the model architecture itself. Learning an identity map is notoriously difficult for neural networks (He et al., 2016; Hardt and Ma, 2017), a point underscored by the fact that simple linear models outperform Video Seal in settings where the architecture should, in principle, excel. Importantly, we do not suggest that naïvely scaling Chunky Seal is a practical path forward. The purpose of this scaling exercise was to explore feasibility, not to advocate for large models in deployment. These results simply illustrate that current architectures fall well short of saturating watermarking capacity, even under generous scaling. Looking ahead, we argue that substantial progress will require new architectural designs, improved losses, and revised training procedures that better encode the inductive biases inherent to watermarking, rather than further scaling of existing models.

We therefore propose a set of sanity checks for the next generation of watermarking methods. A principled approach should scale capacity linearly with image size, decrease capacity linearly with higher PSNR, outperform simple linear or handcrafted baselines, and show predictable drops under stronger augmentations (e.g., $4\times$ lower capacity for a 25% crop). These are necessary for Pareto-optimality and can steer the community toward watermarks with far higher capacity or quality.

Our analysis is not without limitations. We restricted our study to image watermarking, though the insights likely carry over to video. Theoretical bounds are derived only for analytically tractable setups, with some cases relying on numerical integration that becomes impractical at higher resolutions. Our robustness bounds are heuristic rather than formal, leaving ample room for sharper theoretical advances. Finally, while Chunky Seal delivers clear performance gains, its size and latency highlight the need for future architectures that deliver both higher capacities *and* efficiency.

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

## A EXTENDED RELATED WORK

**Classic principled methods.** Early research on image watermarking was dominated by hand-crafted signal processing techniques, grounded in well-understood mathematical and perceptual models. These methods could operate directly in the pixel domain (Van Schyndel et al., 1994; Bas et al., 2002), or in transform domains, most commonly the discrete cosine transform (DCT, Bors and Pitas, 1996; Piva et al., 1997) and the discrete wavelet transform (DWT, Xia et al., 1998; Barni et al., 2001), as well as combinations of the two (Navas et al., 2008; Feng et al., 2010; Zear et al., 2018). A key insight was that perceptually significant frequencies tend to be preserved under transformations (Cox et al., 1997). Other schemes, such as (Ni et al., 2006), introduced perturbations to all pixels with specific values to enable very large payloads. Despite being principled and accompanied by theoretical guarantees, these methods have limited robustness to perturbations and, in some cases, cause noticeable image degradation.

**Deep learning based-watermarking.** With the development of deep learning techniques for computer vision, it was only natural to extend them to image watermarking. Vukotić et al. (2018) proposed adversarially attacking a fixed image extractors, an idea later built upon by Fernandez et al. (2022) and Kishore et al. (2022). A similar iterative approach, albeit lacking robustness, was attempted for steganography by Chen et al. (2023). LISO operates by iteratively optimizing the embedding of information into images, but its performance degrades sharply under even mild compression. Notably, the only robustness-aware variant, LISO-JPEG, achieves merely ∼1 bpp at 19.7 dB PSNR, which is well below practical watermarking quality and is only robust to JPEG compression with CRF 80. This demonstrates that LISO is non-practical for real-world watermarking applications due to its weak robustness and low fidelity.

More popular and successful methods train purpose-built neural networks. Early convolutional models, such as those introduced by Mun et al. (2017) and Zhu et al. (2018), established the feasibility of CNN-based watermarking. Subsequently, architectures based on U-Net (Ronneberger et al., 2015) gained prominence, leveraging multi-resolution representations and residual connections to enable greater network depths. Recent work has also explored watermarking in the latent space of diffusion models Bui et al. (2023b). Advances in perceptual loss design have proven critical: careful loss selection and tuning improved both image quality (Bui et al., 2023a; Fernandez et al., 2024; Xu et al., 2025) and robustness (Tancik et al., 2020; Jia et al., 2021; Pan et al., 2024). Beyond robustness, methods for watermark localization and multi-message embedding have been proposed (Sander et al., 2025; Wang et al., 2024; Zhang et al., 2024), while add-on techniques such as adversarial training (Luo et al., 2020) and attention mechanisms (Zhang et al., 2020) further enhance performance. Although combining image generation with watermarking has also been studied (Fernandez et al., 2023; Wen et al., 2023; Kim et al., 2023; Hong et al., 2024; Ci et al., 2024), such generative approaches are beyond the scope of this work.

**Information-theoretic capacity of watermarking** A natural foundation for analyzing watermarking capacity is the seminal framework of Gel'fand and Pinsker (1980), which characterizes communication over channels with random parameters. In the watermarking context, the *cover image* acts as the channel state: it is known to the encoder (the hider) but not to the decoder. The Gel'fand–Pinsker theorem expresses the capacity of such channels as

$$C = \max_{p(u|s),\, x(u,s)} \big[ I(U;Y) - I(U;S) \big],$$

where $S$ denotes the cover signal, $U$ an auxiliary variable chosen by the encoder, $X$ the watermarked image, and $Y$ the attacked image. This framework underlies nearly all classical information-theoretic analyses of watermarking.

Early works such as Costa (1983) and Cohen and Lapidoth (2002) assume Gaussian cover signals and Gaussian, memoryless attacks. Under these assumptions, the decoder's ignorance of the cover does not reduce capacity: the achievable rate depends only on watermark power and noise power. Quantization-based analyses (Chen and Wornell, 2002) operate under similarly stylized Gaussian models.

More sophisticated approaches were introduced by Moulin and O'Sullivan (2003) and Moulin and Koetter (2005), who cast watermarking as a strategic game between hider and attacker and defined the notion of *hiding capacity*. These formulations characterize the fundamental tradeoff between embedding distortion $D_1$ and attack distortion $D_2$, and have inspired a long line of follow-up work

(e.g., Somekh-Baruch and Merhav, 2004 and Merhav, 2005). However, these analyses remain grounded in Euclidean or Hamming distortion metrics and in pixelwise probabilistic models of the cover and the attack. Even extensions that allow more complex or adversarial behavior—such as pixel permutations—still operate within a per-pixel, additive-noise paradigm.

These classical works provide valuable theoretical insight, but they share the same foundational assumption: watermarking is modeled as a channel coding problem over an idealized Euclidean signal space, with attacks represented as noise in that space. As discussed next, this assumption diverges sharply from the realities of digital images and modern watermark attacks.

**The need for a new framework for watermarking capacity**    The classical Gel'fand and Pinsker (1980) framework treats watermarking as a *per-pixel communication problem*: attacks are modeled as additive or memoryless noise, distortions are measured in Euclidean or Hamming metrics, and images are assumed to be continuous signals with well-defined probability densities. The capacity expression

$$I(U;Y) - I(U;S)$$

reveals the central limitation of this view: mutual information depends on the *absolute probability distribution* of natural images. Such distributions are unknown, representation-dependent (RGB vs. YCbCr, gamma correction, quantization, compression), and fundamentally inestimable. Because $I(U;Y)$ and $I(U;S)$ require these true underlying densities—not empirical or relative statistics—Gel'fand–Pinsker capacities cannot be instantiated for real images.

Moreover, the dominant failure modes of watermarking systems are not pixelwise perturbations but *geometric transformations*: cropping, resampling, rotations, perspective changes, and nonrigid warps that *move* pixels rather than modify their intensities. These structured distortions introduce strong spatial dependencies that fall far outside memoryless noise models and cannot be meaningfully captured by $\ell_2$ or related Euclidean metrics. Real images also inhabit discrete, quantized spaces such as $\{0, \ldots, 255\}^3$, making differential-entropy–based Gaussian capacity formulas inapplicable. As a result, classical information-theoretic capacity predictions are tied to an idealized Euclidean signal model that digital images simply do not satisfy.

Because real attacks are predominantly geometric and because natural-image densities are inaccessible, classical information-theoretic models cannot yield realistic or actionable capacity estimates. They cannot answer practical questions such as:

> *How many bits can be reliably embedded in a* $128 \times 128$ *image while remaining recoverable after rotations, cropping, or other structured geometric distortions?*

Our goal is to develop a capacity theory grounded in the *geometry of images*—one that predicts how many bits can be reliably stored under realistic transformations such as affine and nonrigid warps, and that provides bounds and constructions reflecting the true failure modes of modern watermarking systems.

# B  EQUIVALENCE OF PSNR AND THE $\ell_2$ BALL CONSTRAINT

In the main text we stated that imposing a minimum PSNR $\tau$ is equivalent to requiring the water-marked image to remain within an $\ell_2$ ball around the cover image. Here we give the short derivation and clarify how the radius is defined in terms of both $\ell_2$ and MSE distances.

PSNR is defined from the mean squared error (MSE) between two images:

$$\text{PSNR}(\boldsymbol{x}, \tilde{\boldsymbol{x}}) = 10 \log_{10} \frac{(\texttt{max possible pixel value})^2}{\text{MSE}(\boldsymbol{x}, \tilde{\boldsymbol{x}})}.$$

The MSE measures the *average* squared pixel difference, while the squared $\ell_2$ norm measures the *total* squared difference across all $cwh$ pixels. The two are linked by

$$\text{MSE}(\boldsymbol{x}, \tilde{\boldsymbol{x}}) = \frac{1}{cwh} \|\boldsymbol{x} - \tilde{\boldsymbol{x}}\|_2^2.$$

Thus, a PSNR threshold on MSE can be rephrased as a maximum allowable $\ell_2$ distance between $\boldsymbol{x}$ and $\tilde{\boldsymbol{x}}$. The connection between the two is:

$$\text{PSNR}(\boldsymbol{x}, \tilde{\boldsymbol{x}}) \geq \tau \iff 10 \log_{10}\Big(\frac{\rho^2}{\text{MSE}}\Big) \geq \tau$$

$$\iff \text{MSE} \leq \frac{\rho^2}{10^{\tau/10}}$$

$$\iff \|\boldsymbol{x} - \tilde{\boldsymbol{x}}\|_2^2 \leq cwh \cdot \frac{\rho^2}{10^{\tau/10}}$$

$$\iff \|\boldsymbol{x} - \tilde{\boldsymbol{x}}\|_2 \leq \rho\sqrt{cwh}\, 10^{-\tau/20}.$$

We will define

$$\epsilon(\tau) = \rho\sqrt{cwh}\, 10^{-\tau/20}.$$

In other words, $\epsilon(\tau)$ specifies the largest $\ell_2$ distance (equivalently, the largest total squared pixel error) allowed by the PSNR constraint.

## C  VOLUME-BASED ESTIMATION OF THE NUMBER OF GRID POINTS IN HYPERSPHERES

Calculating watermarking capacity under a PSNR constraint reduces to counting how many valid images (grid points in the pixel space) lie inside an $\ell_2$ ball of radius $\epsilon$ around the cover image. This is equivalent to asking how many integer grid points fall inside such a ball: a problem without a general closed-form solution. In two dimensions this becomes the well-known *Gauss circle problem* (Gauss, 1837; Hardy, 1915). In higher dimensions, and particularly for large radii, the count can be well-approximated by the volume of the $n$-dimensional ball:

$$\text{Vol}(B_n\left[\cdot, r\right]) = \frac{\pi^{n/2}}{\Gamma\left(\frac{n}{2} + 1\right)} r^n. \tag{3}$$

We can approximate the number of integer points inside the ball with its volume. Simply, each grid point corresponds to a unit cube in space, so counting grid points is almost the same as measuring volume. The only difference comes from the boundary of the ball: some cubes are cut by the surface, so they are only partially inside. The absolute error grows as $\mathcal{O}(r^{(n-1)/2})$ (Walfisz, 1957; Mitchell, 1966). Since the volume itself grows as $\mathcal{O}(r^n)$, the relative error decreases as $\mathcal{O}(r^{-(n+1)/2})$. Thus for large radii the volume approximation is quite accurate. For small radii though, this error is significant, as shown in Figure 7, hence we will use exact counting (Appendix D) for these cases.

## D   EXACT COUNTING GRID POINTS IN HYPERSPHERES FOR SMALL RADII

In Section 2.3 we need to calculate how many integer points are in the interior of small $\ell_2$ balls for which approximating the number of integer points with the volume of the ball as in Appendix C is inaccurate. Naively, we can iterate through the points in the smallest hypercube with integer coordinates that contains our ball and check which points would fall inside the ball. See Algorithm 1 for one implementation of this.

---

**Algorithm 1:** Brute Force Count of Lattice Points in a Hypersphere

**1  Function** BruteForceCount(radius : $\mathbb{R}_{\geq 0}$, dim : $\mathbb{Z}^+$)**:**

**2**  $\quad$ rsq $\leftarrow$ radius$^2$;

**3**  $\quad$ single_axis_points $\leftarrow \{-\lfloor \text{radius} \rfloor, \ldots, \lfloor \text{radius} \rfloor\}$;

**4**  $\quad$ counter $\leftarrow 0$;

**5**  $\quad$ **foreach** $p \in$ product(single_axis_points, repeat = dim) **do**

**6**  $\quad\quad$ **if** $\sum_{p=1}^{dim} p_i^2 \leq rsq$ **then**

**7**  $\quad\quad\quad$ counter $\leftarrow$ counter $+ 1$;

**8**  $\quad$ **return** counter;

---

Obviously, Algorithm 1 does not scale beyond very low dimensions because it has complexity that is exponential in the dimension. Luckily, one can leverage symmetries to reduce the number points to be checked. We take the method by (Mitchell, 1966) with pseudo-code presented in Algorithm 2. Note that this can also be further sped up by caching the calls to S.

---

**Algorithm 2:** Count Lattice Points in a Hypersphere (Mitchell's Method)

**1  Function** PointsInHypersphereMitchell(dim : $\mathbb{Z}^+$, radius : $\mathbb{R}_{\geq 0}$)**:**

**2**  $\quad$ **return** S(dim, radius$^2$, $\infty$);

**3  Function** S($m : \mathbb{Z}_{\geq 0}$, $Z : \mathbb{R}$, $J : \mathbb{Z} \cup \{\infty\}$)**:**

**4**  $\quad$ **if** $m = 0$ **then**

**5**  $\quad\quad$ **if** $Z \geq 0$ **then**

**6**  $\quad\quad\quad$ **return** 1;

**7**  $\quad\quad$ **else**

**8**  $\quad\quad\quad$ **return** 0;

**9**  $\quad$ $N \leftarrow \left\lfloor \sqrt{Z/m} \right\rfloor$;

**10**  $\quad$ $r \leftarrow (2N + 1)^m$;

**11**  $\quad$ **for** $i \leftarrow 1$ **to** $m - 1$ **do**

**12**  $\quad\quad$ $\text{MIN}_i \leftarrow \left\lfloor \min\left(\sqrt{Z/i},\ J - 1\right) \right\rfloor$;

**13**  $\quad\quad$ **for** $J_m \leftarrow N + 1$ **to** $\text{MIN}_i$ **do**

**14**  $\quad\quad\quad$ $r \leftarrow r + \binom{m}{i} \cdot 2^i \cdot$ S($m - i$, $Z - iJ_m^2$, $J_m$);

**15**  $\quad$ **return** $r$;

---

Unfortunately, Algorithm 2 is not applicable if we want to compute the number of lattice points in the intersection between the hypersphere and a hypercube, as often needed when computing the number of valid images available for watermarking in the present paper. In such cases, we can use the following simple algorithm which is faster than the naive Algorithm 1 but slower than Algorithm 2. Note that, again, this can be significantly sped up by caching the calls to IterativeCountWithBounds.

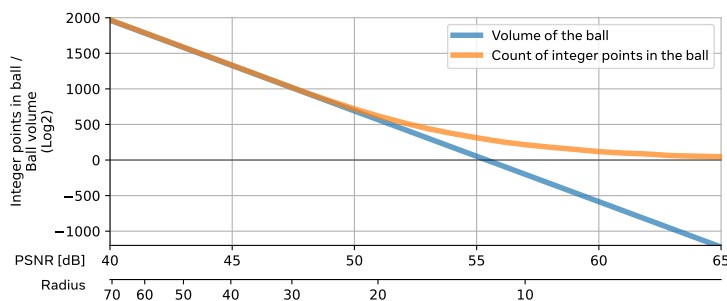

Figure 7: **Discrepancy between the volume of a ball and the number of integer lattice points contained in it for small $\epsilon(\tau)$.** When the radius is small, the volume-based approximation Bound 3 underesimates the actual number of integer points. In such cases we use the exact count Bound 4 instead. Evaluated for a $16 \times 16 \times 3 = 768$-dimensional ball.

---

**Algorithm 3:** Count Lattice Points in a Hypersphere with Bounds

---

**1 Function**
    PointsInHypersphereWithBounds(dim : $\mathbb{Z}^+$, radius : $\mathbb{R}_{\geq 0}$, bounds : $(\mathbb{R} \times \mathbb{R})^{\text{dim}}$)**:**
**2**    int_bounds $\leftarrow (\lceil b_0 \rceil, \lfloor b_1 \rfloor)$ for each $(b_0, b_1)$ in bounds;
**3**    **return** IterativeCountWithBounds($\text{radius}^2$, dim, int_bounds);

**4 Function** IterativeCountWithBounds(rsq : $\mathbb{R}_{\geq 0}$, dim : $\mathbb{Z}_{\geq 0}$, bounds : $(\mathbb{Z} \times \mathbb{Z})^{dim}$)**:**
**5**    **if** dim $= 0$ **then**
**6**        **return** 1 if rsq $\geq 0$, else 0;
**7**    count $\leftarrow 0$;
**8**    $M \leftarrow \lfloor \sqrt{\text{rsq}} \rfloor$;
**9**    lb $\leftarrow \max(-M, \text{bounds}[0][0])$;
**10**    ub $\leftarrow \min(M, \text{bounds}[0][1])$;
**11**    **for** $i \leftarrow$ lb **to** ub **do**
**12**        **if** $i^2 \leq$ rsq **then**
**13**            count $\leftarrow$ count$+$ IterativeCountWithBounds(rsq $- i^2$, dim $- 1$, bounds$[1:]$);
**14**    **return** count;

---

## E  BOUNDS FOR GRAY IMAGE IN THE NON-TRIVIAL INTERSECTION CASE

We expand on the exact bound on the capacity under only PSNR constraint (no robustness) in the non-trivial intersection case, i.e., for medium PSNR values, as discussed in Section 2.3.3. We use the volume-based approximation approach, reducing the problem to finding the volume of an intersection of a hypercube and a hypersphere. Unfortunately, there is no closed-form solution. However, with some care for the numerical precision, we can compute these intersections with numerical integration. First, observe that we can express the volume of the intersection of an arbitrary ball and hypercube as the volume of the intersection of appropriately transformed hypercube with the unit ball:

$$\text{Vol}\left[\prod_{j=1}^{n}[\alpha_j, \beta_j] \cap B_n[\boldsymbol{x}, r]\right] = r^n \, \text{Vol}\left[\prod_{j=1}^{n}\left[\frac{\alpha_j - \boldsymbol{x}_j}{r}, \frac{\beta_j - \boldsymbol{x}_j}{r}\right] \cap B_n[\boldsymbol{0}, 1]\right]. \quad (4)$$

The right-hand side of Equation (4) can be represented as an infinite sum, which, in practice, can be approximated via truncation. We will use the following result originally due Constales (1997) and generalized by Aono and Nguyen (2017, Theorem 4):

**Theorem 1** (Volume of the intersection of a cube and a ball). *Let $S(x) = \int_0^x \sin(t^2)\,dt$ and $C(x) = \int_0^x \cos(t^2)\,dt$ be the Fresnel integrals.[3] Let $\alpha_j \leq \beta_j$ for $1 \leq j \leq n$ and $\ell = \sum_{j=1}^{n}\max(\alpha_j^2, \beta_j^2)$. Then:*

$$\text{Vol}\left[\prod_{j=1}^{n}[\alpha_j, \beta_j] \cap B_n[\boldsymbol{0}, 1]\right] = \begin{cases} K & \text{if } \ell \leq 1 \\ \left(\frac{1}{2} - \frac{\sum_{j=1}^{n}\alpha_j^2 + \beta_j^2 + \alpha_j\beta_j}{3\ell} + \frac{1}{\ell} + \frac{1}{\pi}\,\text{Im}\sum_{k=1}^{\infty}\frac{\Phi(-2\pi k/\ell)}{k}e^{2i\pi k/\ell}\right)K & \text{if } \ell > 1 \end{cases}$$

*where $K = \prod_{j=1}^{n}|\beta_j - \alpha_j|$ and $\Phi$ is defined as*

$$\Phi(\omega) = \prod_{j=1}^{n} \frac{\left(C(\beta_j\sqrt{|\omega|}) - C(\alpha_j\sqrt{|\omega|})\right) + i\,\text{sign}(\omega)\left(S(\beta_j\sqrt{|\omega|}) - S(\alpha_j\sqrt{|\omega|})\right)}{(\beta_j - \alpha_j)\sqrt{|\omega|}}.$$

A number of speed-ups and numerical performance optimization tricks can be used when computing the terms in Theorem 1. For example, when all $\alpha_j = \alpha_1$ and $\beta_j = \beta_1$ for all $1 \leq j \leq n$ and when $\alpha_j = -\beta_j$, which is often the case in the setups we consider. We provide such optimized implementation in Algorithm 4.

Theorem 1 directly gives us a bound on the capacity for the non-trivial intersection case:

**Bound 5: Gray image, PSNR constraint (medium PSNR, numerical integration).** The capacity of a gray image $\boldsymbol{x}_g$ under a minimum PSNR constraint $\tau$ in the ambient space $\mathcal{A}$ is upper-bounded by

$$\texttt{capacity[in bits]} \approx \log_2\left[\epsilon^{cwh}\,\text{Vol}\left[\prod_{j=1}^{cwh}\left[-\frac{\rho/2}{\epsilon(\tau)}, \frac{\rho/2}{\epsilon(\tau)}\right] \cap B_{cwh}[\boldsymbol{0}, 1]\right]\right]$$

$$= cwh\log_2\epsilon(\tau) + \log_2\text{Vol}\left[\prod_{j=1}^{cwh}\left[-\frac{\rho/2}{\epsilon(\tau)}, \frac{\rho/2}{\epsilon(\tau)}\right] \cap B_{cwh}[\boldsymbol{0}, 1]\right],$$

with the volume computed by truncating the sum in Theorem 1.
**Bound validity:** If $B_{cwh}[\boldsymbol{x}_g, \epsilon(\tau)] \not\subset C_\mathcal{I}$ and $C_\mathcal{I} \not\subset B_{cwh}[\boldsymbol{x}_g, \epsilon(\tau)]$, which happens when $\rho/2 \leq \epsilon(\tau) \leq \rho/2\sqrt{cwh}$ or, equivalently from Equation (1), when $20\log_{10}2 \leq \tau \leq 20\log_{10}(2\sqrt{cwh})$. Assuming $cwh$ not too large (otherwise numerical evaluation of the bound becomes intractable).

The main limitation of Bound 5 is that it becomes computationally intractable to compute it for large $cwh$. In our implementation, we could evaluate it up to $cwh$ of several hundred. However, as can be seen in Figure 3 left, Bound 5 can be well-approximated by simply considering the minimum of Bound 2 and Bound 3. Therefore, for large resolutions we will use Bound 6.

---

[3]Note that some software libraries provide alternative (normalized) definitions for the Fresnel integrals: $\tilde{S}(x) = \int_0^x \sin(\pi x^2/2)$ and $\tilde{C}(x) = \int_0^x \cos(\pi x^2/2)$. The two are related as such: $S(x) = \sqrt{\pi/2}\,\tilde{S}(\sqrt{2/\pi}x)$ and $C(x) = \sqrt{\pi/2}\,\tilde{C}(\sqrt{2/\pi}x)$.

**Algorithm 4:** Computes the volume of the intersection between a hypersphere and a box

**1 Function** ApplyWithGrouping($f$ : function, args, mode : (product, sum)):

   // Applies a function $f$ to arguments, grouping them for stability.

  **2**   pairs $\leftarrow$ list(zip($*$args));

  **3**   $\mathcal{C} \leftarrow$ Counter(pairs) ;                     // Count unique argument tuples

  **4**   **if** mode = product **then**

  **5**     |  result $\leftarrow 1$;

  **6**   **else if** mode = sum **then**

  **7**     |  result $\leftarrow 0$;

  **8**   **foreach** $p \in \mathcal{C}$.keys() **do**

  **9**     |  $c \leftarrow \mathcal{C}[p]$;

  **10**    |  $r \leftarrow f(p)$;

  **11**    |  **if** mode = product **then**

  **12**    |    |  result $\leftarrow$ result $\cdot r^c$;

  **13**    |  **else if** mode = sum **then**

  **14**    |    |  result $\leftarrow$ result $+ c \cdot r$;

  **15**   **return** result;

**16 Function** $\Phi_{\text{inner}}(\alpha : \mathbb{R}, \beta : \mathbb{R}, \omega : \mathbb{R})$:

  **17**   $\omega' \leftarrow \sqrt{|\omega|}$;

  **18**   **if** $\alpha = -\beta$ **then**

  **19**    |  $T_1 \leftarrow 2 \cdot$ fresnelc$(\beta\omega')$;

  **20**    |  $T_2 \leftarrow i \cdot \text{sgn}(\omega) \cdot 2 \cdot$ fresnels$(\beta\omega')$;

  **21**   **else**

  **22**    |  $T_1 \leftarrow$ fresnelc$(\beta\omega') -$ fresnelc$(\alpha\omega')$;

  **23**    |  $T_2 \leftarrow i \cdot \text{sgn}(\omega) \cdot ($fresnels$(\beta\sqrt{\omega_{abs}}) -$ fresnels$(\alpha\omega'))$;

  **24**   **return** $(T_1 + T_2)/((\beta - \alpha)\omega')$;

**25 Function** $\Phi(\omega : \mathbb{R}, \boldsymbol{\alpha}, \boldsymbol{\beta})$:

  **26**   $f_{\text{inner}} \leftarrow ((\alpha, \beta) \mapsto \Phi_{\text{inner}}(\alpha, \beta, \omega) )$;

  **27**   **return** ApplyWithGrouping$(f_{inner}, [\boldsymbol{\alpha}, \boldsymbol{\beta}], \text{product})$;

**28 Function** BallCubeIntersection($R : \mathbb{R}_{>0}, \boldsymbol{\alpha}, \boldsymbol{\beta}, N_{sum} : \mathbb{Z}^+$):

  // Trim scaled bounds to the unit ball range $[-1, 1]$

  **29**   $\boldsymbol{\alpha}' \leftarrow$ elementwise_max$(-1, \boldsymbol{\alpha}/R)$;

  **30**   $\boldsymbol{\beta}' \leftarrow$ elementwise_min$(1, \boldsymbol{\beta}/R)$;

  **31**   $\ell \leftarrow$ ApplyWithGrouping$((a, b) \mapsto (\max(-a, b))^2, [\boldsymbol{\alpha}', \boldsymbol{\beta}'], \text{sum})$;

  **32**   $E_X \leftarrow {}^1/_3$ ApplyWithGrouping$((a, b) \mapsto a^2 + b^2 + ab, [\boldsymbol{\alpha}', \boldsymbol{\beta}'], \text{sum})$;

  **33**   $V_{\text{scale}} \leftarrow$ ApplyWithGrouping$((a, b) \mapsto \log_2(b - a), [\boldsymbol{\alpha}', \boldsymbol{\beta}'], \text{sum})$;

  **34**   $T_1 \leftarrow {}^1/_2$;    $T_2 \leftarrow {}^{E_X}/_\ell$;    $T_3 \leftarrow {}^1/_\ell$;

  **35**   $S_4 \leftarrow 0$;

  **36**   **for** $k \leftarrow 1$ **to** $N_{sum}$ **do**

  **37**    |  $\omega_k \leftarrow -2\pi k/\ell$;

  **38**    |  $S_4 \leftarrow S_4 + \frac{1}{k} \cdot \Phi(\omega_k, \boldsymbol{\alpha}', \boldsymbol{\beta}') \cdot$ expjpi$(2k/\ell)$;

  **39**   $T_4 \leftarrow$ Im$(S_4)/\pi$;

  **40**   $V_{\text{norm}} \leftarrow T_1 - T_2 + T_3 + T_4$;

  **41**   **return** $\log_2(V_{\text{norm}}) + V_{\text{scale}} +$ len$(\boldsymbol{\alpha}') \cdot \log_2(R)$;

## F  BOUNDS FOR ARBITRARY COVER IMAGES

In Section 2.4 we extended the analysis from centred covers to arbitrary images. Here we go into detail about how the corresponding bounds were derived.

When pixels are saturated, the cover may lie on the boundary of the cube of cube $C_{\mathcal{I}}$. The most adverse case is when all pixels are saturated, i.e., the cover at a corner. This minimizes the overlap between the PSNR ball and the grid and hence provides a lower bound on capacity for any image.

By symmetry, the ball is evenly divided among the $2^{cwh}$ orthants of $\mathbb{R}^{cwh}$, so only a fraction $1/2^{cwh}$ of its volume lies within the grid. In two dimensions this corresponds to a quarter of a circle inside a square corner, and in three dimensions to one eighth of a sphere inside a cube corner. Although this seems drastic, the effect on capacity is limited: at most $cwh$ bits, i.e. one bit per pixel.

We now provide the detailed derivations of Bounds 7 to 9, which are the analogues of the centered bounds from Section 2.3, adapted to the corner case.

**Bound 7: Arbitrary image, PSNR constraint (high PSNR, volume approximation, analogous to Bound 3).** The capacity of an image $x$ under a minimum PSNR constraint $\tau$ in the ambient space $\mathcal{A}$ is upper-bounded by

$$\texttt{capacity[in bits]} \approx \log_2 \left\lceil \frac{\operatorname{Vol} B_{cwh}\left[\cdot, \epsilon(\tau)\right]}{2^{cwh}} \right\rceil$$

$$= \frac{cwh}{2} \log_2 \pi + cwh \log_2 \epsilon(\tau) - \frac{\ln \Gamma\left(\frac{cwh}{2} + 1\right)}{\ln 2} - cwh.$$

**Bound validity:** If $\epsilon(\tau) \leq \rho/2$ or, equivalently from Equation (1), if $\tau \geq 20 \log_{10}(2\sqrt{cwh})$. Assuming $\epsilon(\tau)$ not too small for the volume approximation to be valid (see Bound 8 for small $\epsilon(\tau)$).

**Bound 8: Arbitrary image, PSNR constraint (high PSNR, exact count, analogous to Bound 4).** The capacity of an image $x$ under a minimum PSNR constraint $\tau$ in the ambient space $\mathcal{A}$ is upper-bounded by

$$\texttt{capacity[in bits]} \approx \log_2 \texttt{PointsInHypersphereWithBounds}(cwh, \epsilon(\tau), (0, 2^k)),$$

with `PointsInHypersphereWithBounds` defined in Algorithm 3.
**Bound validity:** If $\epsilon(\tau) \leq \rho/2$ or, equivalently from Equation (1), if $\tau \geq 20 \log_{10}(2\sqrt{cwh})$. Assuming $\epsilon$ small (otherwise the numerical evaluation becomes intractable, see Bound 7 for large $\epsilon$).

Note that Bound 7 slightly *under-approximates* capacity, since only half of the lattice points lying on the faces of the hypercube are captured by the volume approximation. This explains the discontinuity between Bound 7 and Bound 8 visible in Figure 3 right.

For the other two cases: non-trivial intersection and the cube being fully in the ball, we can directly apply Theorem 1 with appropriate change of bounds. We simply need to adjust the bounds in Bound 5 from $[-\rho/2, \rho/2]$ to $[0, \rho]$:

**Bound 9: Arbitrary image, PSNR constraint (medium PSNR, numerical integration, analogous to Bound 5).** The capacity of an image $x$ under a minimum PSNR constraint $\tau$ in the ambient space $\mathcal{A}$ is upper-bounded by

$$\mathtt{capacity[in\ bits]} \approx \log_2 \left[ \epsilon(\tau)^{cwh} \ \mathrm{Vol}\Big( \prod_{j=1}^{cwh} \Big[0, \tfrac{2^k-1}{\epsilon(\tau)}\Big] \cap B_{cwh}\left[\mathbf{0}, 1\right] \Big) \right]$$

$$= cwh \log_2 \epsilon + \log_2 \mathrm{Vol}\Big( \prod_{j=1}^{cwh} \Big[0, \tfrac{2^k-1}{\epsilon(\tau)}\Big] \cap B_{cwh}\left[\mathbf{0}, 1\right] \Big),$$

with the volume computed by truncating the sum in Theorem 1.

**Bound validity:** If $\epsilon(\tau) \geq \rho/2$ or, equivalently from Equation (1), when $\tau \leq 20 \log_{10}(2\sqrt{cwh})$. Assuming $cwh$ not too large (otherwise the numerical evaluation becomes intractable).

Unlike the centred case, here the symmetry condition $\alpha_j = -\beta_j$ no longer holds, so the simplifications of Theorem 1 cannot be applied. Bound 9 is therefore numerically stable only at relatively low resolutions. Nevertheless, the per-pixel capacity (bpp) is resolution-invariant, so we compute these bounds at $16{\times}16$px.

Bounds 7 to 9 extend the centred-case analysis to arbitrary cover images. Across all three regimes, the penalty of being at the corner of the grid is at most one bit per pixel. Thus, the observed gap between theoretical capacity and practical watermarking performance cannot be explained by image position within the grid.

# G   BOUNDS FOR CAPACITY UNDER ROBUSTNESS CONSTRAINTS

In practice, watermarking requires balancing perceptual quality with robustness. That is, minor modifications to the watermarked image should not prevent the watermark from being extractable. Such modifications might arise in the normal processing of the image (a social media website might compress uploaded images) or might be malicious (to strip provenance information). Typically, one considers a set of transformations against which a watermarking method should be robust.

Robustness comes at a cost: it reduces capacity. To quantify this trade-off we study how robustness constraints reduce the number of images which we can use for watermarking, quantifying the corresponding reduction in capacity. We will focus on robustness to linear transformations, which, though seemingly restrictive, covers or approximates most practical transformations. Linear transformations are also compositional, simplifying the creation of complex transformations from basic ones. Standard augmentation can be directly represented as linear transformations. Appendix H shows how to represent colour space changes, rotation, flipping, cropping and rescaling, as well as a number of intermediate operators that can be used to construct the linear operators corresponding to other transformations.

## G.1   ROBUSTNESS TO LINEAR TRANSFORMATIONS

A linear transformation applied to the ball $B_{cwh}[\mathbf{0}, \epsilon]$ of possible watermarked images can turn it into an ellipsoid (if the transformation has more than one unique singular value) can scale it (if they are all the same but not 1) and can also project it into a lower-dimensional space if the transformation is not invertible. Let's take a linear transformation $M \in \mathbb{R}^{cwh \times cwh}$ that maps an image $\boldsymbol{x}$ to a transformed image $M\boldsymbol{x}$. Note that we further need to apply a quantization operation to map the pixel values of $M\boldsymbol{x}$ to the valid images $\mathcal{I}$. Hence, we have the final transformed image $\boldsymbol{x}' = \mathsf{Q}[M\boldsymbol{x}]$ with $\mathsf{Q}$ being a quantization operator, typically an element-wise rounding or floor operation. The main complication in this setup is that $\mathsf{Q}$ is non-linear. To establish the capacity under a linear transformation, we need to find the subset of the possible watermarked images under only the PSNR constraint that map to unique valid images after applying $M$ and $\mathsf{Q}$ to them.

## G.2   HEURISTIC BOUNDS

A simple approach is to consider the volumetric approach for calculating capacity that we used for Bounds 3, 5, 7 and 9. A linear operator $M$ would change the volume of $B_{cwh}[\mathbf{0}, \epsilon]$ by a factor of $\det M = \lambda_M^1 \times \cdots \times \lambda_M^{cwh}$ (the product of $M$'s eigenvalues) if $M$ is not singular. Note that if $\det M > 0$, then we can also express it as $\det M = \sigma_M^1 \times \cdots \times \sigma_M^{cwh}$, the product of singular values of $M$. If $M$ is singular, then $MB_{cwh}[\mathbf{0}, \epsilon]$ has 0 $cwh$-dimensional volume as some eigenvalues (singular values) would be 0. However, it will have a non-zero $\operatorname{pdet} M = \prod \left\{ \lambda_M^i \mid \lambda_M^i \neq 0, \ i = 1, \ldots, cwh \right\}$ ($\operatorname{rank} M$)-dimensional volume, with $\operatorname{pdet} M$ being the *pseudo-determinant* of $M$.

The change in volume governs the reduction in capacity. However, it is not as simple as just calculating this volume and taking that to be the capacity because of the quantizer $\mathsf{Q}$. Take for example

$$M = \begin{bmatrix} 2 & 0 \\ 0 & 1/2 \end{bmatrix}. \tag{5}$$

The determinant is $\det M = 1$ indicating no volume change and, thus, if we ignore the quantization, no capacity change. However, one of the dimensions is squeezed by a factor of 2, hence we should lose about half of the capacity along this axis. The other axis, that is stretched by a factor of 2, does not create capacity because pairs of these points would have the same preimage. Therefore, assuming we are far from the boundaries of the cube, we should see half the original capacity, if we want to have robustness to this augmentation. Following this observation, we provide an heuristic for the reduction $\xi$ of capacity due to the linear operator and quantization:

$$\xi_M = \prod_{\sigma_M^i \in \Sigma_M : \sigma_M^i > 0} \min(\sigma_M^i, 1), \tag{6}$$

where $\Sigma_M$ are the singular values of $M$. $\xi_M$ captures the combined effect of all singular values of $M$. Each $\sigma_i < 1$ represents a compression that reduces capacity proportionally due to the quantization, while $\sigma_i > 1$ is capped at 1 since stretching cannot create capacity. The product accounts for all $\operatorname{rank} M$ dimensions, so $\xi_M$ reflects the total fraction of capacity that remains after accounting for all reductions.

If, for a moment, we ignore that we need to clip the pixel values in their valid range, we get the following capacity under an *invertible* linear transformation $M$ for $\epsilon$ large enough so that the volume-based approximation is applicable:

$$\texttt{capacity under } M[\text{in bits}] = \texttt{capacity}[\text{in bits}] + \log_2 \xi_M$$
$$\approx \log_2 \mathrm{Vol}\, B_{cwh}\left[\cdot, \epsilon(\tau)\right] + \log_2 \xi_M.$$

If $M$ is singular, however, then we need to compute the capacity under the lower-dimensional projection of $B_{cwh}\left[\cdot, \epsilon(\tau)\right]$ in order to account for the collapsed dimensions:

$$\texttt{capacity under } M[\text{in bits}] \approx \log_2 \mathrm{Vol}\, B_{\mathrm{rank}\, M}\left[\cdot, \epsilon(\tau)\right] + \log_2 \xi_M.$$

Note that the radius $\epsilon(\tau)$ is still computed in the ambient $cwh$-dimensional space.

A further complication caused by the $\sigma_i > 1$ singular values is the possibility of the sphere going out of the bounds of the cube after the transform. Looking again at the example in Equation (5), the stretching along the first dimension might result in some of the watermarked images being clipped and hence mapped to the same image after applying the transformation. We can factor this in by adjusting the bounds corresponding to the cube boundaries in Bounds 3, 4 and 5 accordingly. This results in the heuristic bounds in this section.

First, let's look at the setting where the ball is fully inside the cube *before and after* the transformation, and its radius $\epsilon(\tau)$ is large enough for us to approximate the number of images in it with its volume. Taking into account $M$ being possibly singular, we have:

**Bound 10: Gray image, Linear transformation (Heuristic), PSNR constraint (high PSNR, volume approximation).** The capacity of a gray image $x_g$ under a low minimum PSNR constraint $\tau$ and a linear transformation $M \in \mathbb{R}^{cwh \times cwh}$ in the ambient space $\mathcal{A}$ is upper-bounded as such:

$$\texttt{capacity}[\text{in bits}] \approx \log_2 \xi_M \;\; \mathrm{Vol}\, B_{\mathrm{rank}\, M}\left[\cdot, \epsilon(\tau)\right]$$
$$= \log_2 \xi_M \; \frac{\pi^{(\mathrm{rank}\, M)/2}\, \epsilon(\tau)^{\mathrm{rank}\, M}}{\Gamma\left(\frac{\mathrm{rank}\, M}{2} + 1\right)}$$
$$= \frac{\mathrm{rank}\, M}{2} \log_2 \pi + (\mathrm{rank}\, M) \log_2 \epsilon - \frac{\ln \Gamma\left(\frac{\mathrm{rank}\, M}{2} + 1\right)}{\ln 2} + \sum_{\sigma_M^i \in \Sigma_M : \sigma_M^i > 0} \min(\log_2 \sigma_M^i, 0).$$

**Bound validity:** If $\epsilon(\tau) \leq \rho/2\mu$ with $\mu = \max\{\sigma_1, \ldots, \sigma_{cwh}, 1\}$ or, equivalently when $\tau \geq 20 \log_{10}(2\mu\sqrt{cwh})$. Assuming $\epsilon(\tau)$ not too small (see Bound 11 for small $\epsilon(\tau)$).

Similarly to Bounds 4 and 8, when the radius $\epsilon$ is too small, the volume approximation is poor and we resort to exact counting instead:

**Bound 11: Gray image, Linear transformation (Heuristic), PSNR constraint (high PSNR, exact count).** The capacity of a gray image $x_g$ under a low minimum PSNR constraint $\tau$ and a linear transformation $M \in \mathbb{R}^{cwh \times cwh}$ in the ambient space $\mathcal{A}$ is upper-bounded as such:

$$\texttt{capacity}[\text{in bits}] \approx \log_2 \xi_M \; \texttt{PointsInHypersphereMitchell}\,(\mathrm{rank}\, M, \epsilon)$$
$$= \log_2 \texttt{PointsInHypersphereMitchell}\,(\mathrm{rank}\, M, \epsilon) + \sum_{\sigma_M^i \in \Sigma_M : \sigma_M^i > 0} \min(\log_2 \sigma_M^i, 0)$$

with $\texttt{PointsInHypersphereMitchell}$ as described in Algorithm 2.
**Bound validity:** If $\epsilon(\tau) \leq \rho/2\mu$ with $\mu = \max\{\sigma_1, \ldots, \sigma_{cwh}, 1\}$ or, equivalently when $\tau \geq 20 \log_{10}(2\mu\sqrt{cwh})$. Assuming $\epsilon(\tau)$ is small (otherwise, computationally intractable, see Bound 10 for large $\epsilon(\tau)$).

And finally, we have the non-trivial intersection case, where we need to account for the clipping of the watermarked images to $C_{\mathcal{I}}$, the cube of all possible images:

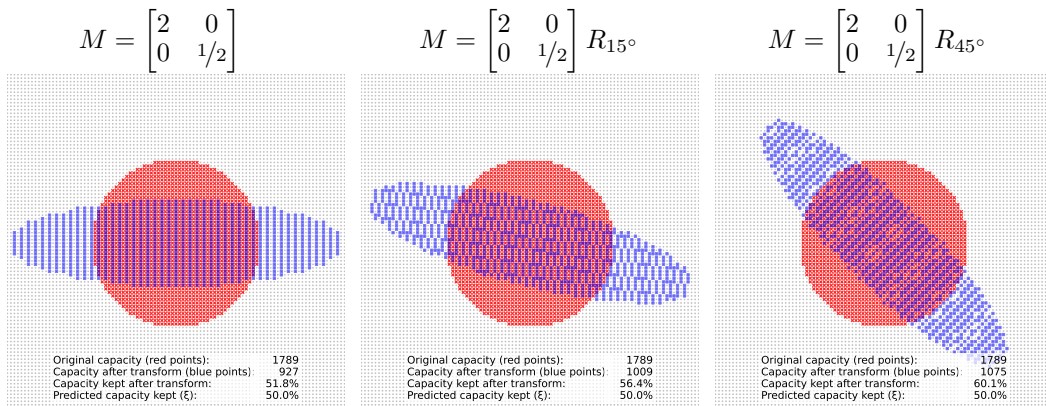

Figure 8: **Quantization can result in higher capacities than predicted by Equation (6).** Showing how rotations of a matrix with singular values smaller and larger than 1 can affect the capacity of a linear transform due to the quantization effects. Despite Equation (6) predicting factor $\xi = 0.5$ for all three cases, for this $M$ we observe larger factors of up to 0.60 in the case of $45°$ rotation.

---

**Bound 12: Gray image, Linear transformation (Heuristic), PSNR constraint (medium PSNR, numerical integration).** The capacity of a gray image $\boldsymbol{x}_g$ under a low minimum PSNR constraint $\tau$ and a linear transformation $M \in \mathbb{R}^{cwh \times cwh}$ in the ambient space $\mathcal{A}$ is upper-bounded as such:

$$\text{capacity[in bits]} \approx \log_2 \left[ \xi_M \, \epsilon^{\text{rank } M} \text{Vol} \left[ \prod_{\sigma_M^i \in \Sigma_M : \sigma_M^i > 0} \left[ -\frac{\rho}{2\epsilon \max(1, \sigma_M^i)}, \frac{\rho}{2\epsilon \max(1, \sigma_i)} \right] \cap B_{\text{rank } M} [\boldsymbol{0}, 1] \right] \right]$$

$$= (\text{rank } M) \log_2 \epsilon(\tau)$$

$$+ \log_2 \text{Vol} \left[ \prod_{\sigma_M^i \in \Sigma_M : \sigma_M^i > 0} \left[ -\frac{\rho}{2\epsilon(\tau) \max(1, \sigma_M^i)}, \frac{\rho}{2\epsilon(\tau) \max(1, \sigma_M^i)} \right] \cap B_{\text{rank } M} [\boldsymbol{0}, 1] \right]$$

$$+ \sum_{\sigma_M^i \in \Sigma_M : \sigma_M^i > 0} \min(\log_2 \sigma_M^i, 0),$$

with the volume computed by truncating the sum in Theorem 1.
**Bound validity:** If $\epsilon(\tau) > \rho/2\mu$ with $\mu = \max\{\sigma_1, \ldots, \sigma_{cwh}, 1\}$ or, equivalently when $\tau \leq 20 \log_{10}(2\mu\sqrt{cwh})$. Assuming rank $M$ not too large (otherwise numerical evaluation of the bound becomes intractable).

We would like to stress that Bounds 10 to 12 are *heuristics* and are near-exact only for axis-aligned transformations, i.e., where $M$ has at most one non-zero value in each row or column.

Higher capacities than predicted by Bounds 10 to 12 are possible. Take $M_\theta = M R_\theta$ as in Equation (5) but multiplied with a rotation matrix $R_\theta$:

$$R_\theta = \begin{bmatrix} \cos\theta & -\sin\theta \\ \sin\theta & \cos\theta \end{bmatrix}.$$

The rotation does not affect the singular values hence the scaling factor $\xi_M = \xi_{M_\theta}$ and Bounds 10 to 12 are unaffected. Nevertheless, the exact count of images that remain after $M_\theta$ and $Q$ (and thus the capacity) is much higher in the rotated case. Figure 8 shows that while $\xi_{M_\theta}$ is 0.5 for all $\theta$, the actual capacity factor at for $\theta = 45°$ is 0.6, i.e., 20% higher. Thus Bounds 10 to 12 are not upper bounds on the capacity under a linear transformation.

Unfortunately, Bounds 10 to 12 are also not lower bounds on the capacity. Take the case of transforming with just $R_\theta$. $R_\theta$ has a determinant 1 for all $\theta$, hence the scaling factor $\xi_{R_\theta}$ is also 1 and therefore, the capacity should be unchanged. However, as can be seen in Figure 9, the empirical $\hat{\xi}_{R_{45°}}$ is just 0.837 when we rotate the disk by $\theta = 45°$. While this particular case has been studied analytically by Vladimirov (2015, Theorem 19) who demonstrated that for large enough disks this reduction is $\hat{\xi}_{R_\theta} = 1 - (\cos\theta + \sin\theta - 1)^2$, convenient results for general linear operators $M$ are unlikely to be possible.

$$M = \begin{bmatrix} 1 & 0 \\ 0 & 1 \end{bmatrix} \qquad M = \begin{bmatrix} 1 & 0 \\ 0 & 1 \end{bmatrix} R_{15°} \qquad M = \begin{bmatrix} 1 & 0 \\ 0 & 1 \end{bmatrix} R_{45°}$$

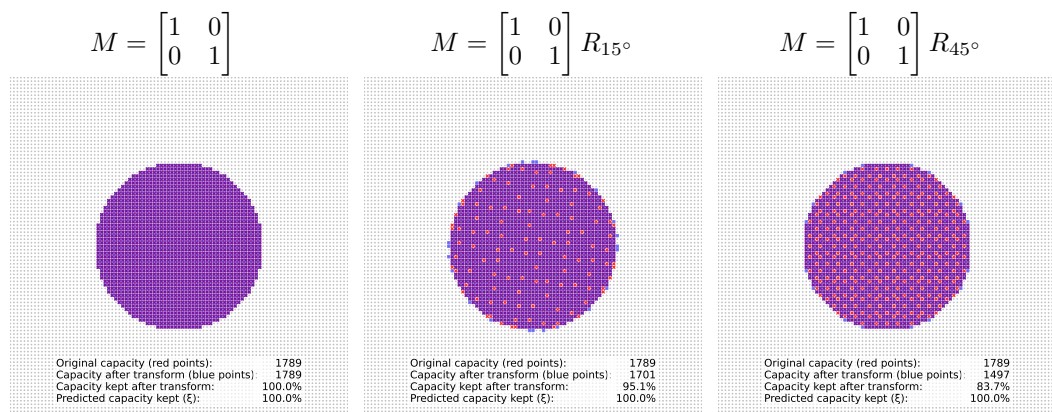

| | |
|---|---|
| Original capacity (red points): | 1789 |
| Capacity after transform (blue points): | 1789 |
| Capacity kept after transform: | 100.0% |
| Predicted capacity kept ($\xi$): | 100.0% |

| | |
|---|---|
| Original capacity (red points): | 1789 |
| Capacity after transform (blue points): | 1701 |
| Capacity kept after transform: | 95.1% |
| Predicted capacity kept ($\xi$): | 100.0% |

| | |
|---|---|
| Original capacity (red points): | 1789 |
| Capacity after transform (blue points): | 1497 |
| Capacity kept after transform: | 83.7% |
| Predicted capacity kept ($\xi$): | 100.0% |

Figure 9: **Quantization can result in lower capacities than predicted by Equation (6).** Showing how rotations of a disk can affect the capacity of a linear transform due to the quantization effects. Despite Equation (6) predicting factor $\xi = 1$ for all three cases, for rotations we observe larger factors of as little as 0.837 in the case of $45°$ rotation.

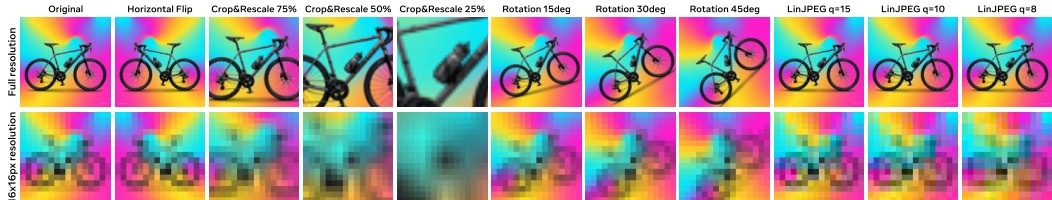

Figure 10: **Robustness constraints considered in this paper.** The figure illustrates the linear transformations we evaluate: Horizontal Flip, Crop&Rescale, Rotation and Linearized JPEG compression. We show them both at full resolution and at $16 \times 16$px at which we compute the bounds in Section 2.5.

Nevertheless, for practical purposes, we believe that Bounds 10 to 12 are mostly lower bounds. For instance, when we have both "squish" and "stretch" axes, i.e., singular values both smaller and larger than 1 —which is the case for the transformations we consider— then the true capacity can be much higher than predicted due to the rounding operation filling the space better as seen in Figure 8. Therefore, in general, we would consider this bound to be a good heuristic.

To evaluate the effect of Bounds 10 to 12 on the watermarking capacity, we considered horizontal flipping, cropping and rescaling, rotation (around the centre of the image), as illustrated in Figure 10. The construction of the respective matrices is described in Appendices H.3.3, H.3.5 and H.3.6. As can be seen in Figure 11 they all, except for the horizontal flip, have singular values above and below 1 and are rank deficient. Figure 4 has Bounds 10 to 12 for the Crop&Rescale and the Rotation transformations and compares them with the bounds without robustness constraints from Section 2.3. Rotations have a roughly $2$ bpp decrease in capacity mostly driven by the loss of

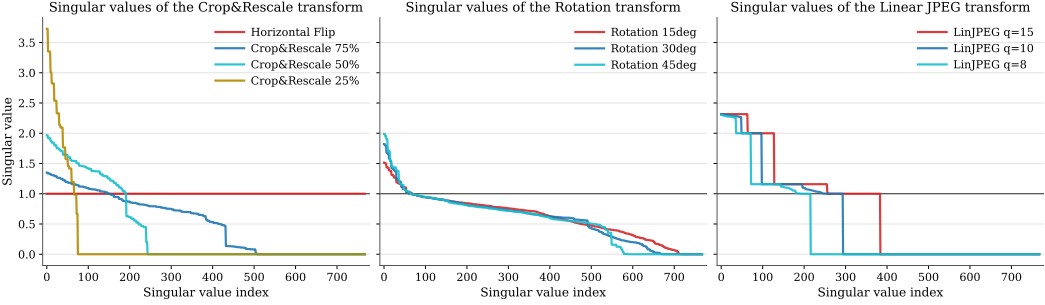

Figure 11: **Singular values of the linear transformations.** Plotted are the singular values of the linear transformations, sorted in descending order.

capacity at the corners and the effects of the interpolation. As expected, aggressive crops reduce the capacity significantly. At $40\,\text{dB}$, cropping to $0.25$ of the image results in about $0.5\,\text{bpp}$, down from more than $3\,\text{bpp}$ without the robustness constraint. Still, $0.5\,\text{bpp}$ implies capacity of $98,304$ bits for a $256\times256\text{px}$ image, considerably larger capacity than what we observe in practice. Therefore, robustness to augmentations does not seem to explain the much lower capacities that we see in practice.

### G.3 Conservative bounds

While we are confident in the heuristic Bounds 10 to 12, it is nevertheless possible that, in practice, capacities suffer from a similar problem as the rotation $R_\theta$ (Figure 9) and hence are lower than predicted by the heuristic bounds. Unfortunately, due to the non-linear nature of the quantizer $\mathsf{Q}$ and the curse of dimensionality, providing an actual lower bound for $\xi_M$ proves difficult. Still, we outline one approach here which, while extremely conservative, is a valid lower bound.

The idea of this conservative bound is to find an upper bound for the cardinality of the preimage of any transformed image, that is, an upper bound to how many images would be mapped to the same transformed image under $M$ and $\mathsf{Q}$. Every untransformed point (image) $\boldsymbol{x}$ in $C_\mathcal{I} \cap B_{cwh}\left[\boldsymbol{x}_g, \epsilon\right] \cap \mathbb{Z}^{cwh}$ maps to a point $\boldsymbol{x}'$ in $C_\mathcal{I} \cap \mathbb{Z}^{cwh}$ after being transformed by $\mathsf{Q}[M\boldsymbol{x}]$. However, multiple images can map to the same image by the transformation, i.e., $|M^+\mathsf{Q}^{-1}[\boldsymbol{x}'] \cap C_\mathcal{I} \cap B_{cwh}\left[\boldsymbol{x}_g, \epsilon\right] \cap \mathbb{Z}^{cwh}|$ might be more than 1. Here $M^+$ is the (Moore–Penrose) pseudo-inverse of $M$ and $\mathsf{Q}^{-1}[\boldsymbol{x}']$ is the set of points that $\mathsf{Q}$ maps to $\boldsymbol{x}'$, thus $M^+\mathsf{Q}^{-1}[\boldsymbol{x}']$ are all points $\boldsymbol{x}$ such that $\mathsf{Q}[M\boldsymbol{x}] = \boldsymbol{x}'$. Define $N$ to be the number of points available for watermarking without the robustness constraint (i.e., $2^{\text{capacity[in bits]}}$) and $n$ to be the number of unique images that are robust to the transformation $\mathsf{Q}M$:

$$N = \left|C_\mathcal{I} \cap B_{cwh}\left[\boldsymbol{x}_g, \epsilon\right] \cap \mathbb{Z}^{cwh}\right|,$$

$$n = \left|\left\{\boldsymbol{x}' \in C_\mathcal{I} \cap \mathbb{Z}^{cwh} \mid \exists \boldsymbol{x} \in C_\mathcal{I} \cap B_{cwh}\left[\boldsymbol{x}_g, \epsilon\right] \cap \mathbb{Z}^{cwh} \text{ such that } \boldsymbol{x}' = \mathsf{Q}[M\boldsymbol{x}]\right\}\right|. \quad (7)$$

In other words, $n$ is the capacity that we can achieve while being robust to the linear transformation $M$. Note that the scaling factor $\xi_M$ is precisely $n/N$. Section 2.3 was concerned with finding $N$, here we will provide an upper bound to $n$. While obtaining $n$ directly is difficult, if we know that every transformed point has at most $K$ preimage points, i.e.:

$$|M^+\mathsf{Q}^{-1}[\boldsymbol{x}] \cap C_\mathcal{I} \cap B_{cwh}\left[\boldsymbol{x}_g, \epsilon(\tau)\right] \cap \mathbb{Z}^{cwh}| \le K, \; \forall \boldsymbol{x}$$

then we know that $n \ge \lceil N/K \rceil$. In other words we have that the scaling factor must be lower-bounded as $\xi_M \ge 1/K$. So the problem of finding a lower bound to $\xi_M$ has reduced to obtaining $K$, an upper bound to the number of preimage points that any transformed image can have.

The quantization operation $\mathsf{Q}$ maps a hypercube of side 1 to a given image, regardless of whether the rounding or floor quantization is used. The maximum number of points that can fit in the preimage under $M$ of such a unit cube is the $K$ we are looking for. The preimage of a hypercube under a linear operation $M$, when restricted to a hypersphere of radius $\epsilon$, can be over-approximated with a zonotope (with proof in Appendix I):

**Theorem 2.** *Given a hypercube $C = \boldsymbol{b} + [0,1]^n \subset \mathbb{R}^n$, $\boldsymbol{b} \in \mathbb{R}^n$ and a possibly singular matrix $M \in \mathbb{R}^{n \times n}$ giving rise to the map $f_M(\boldsymbol{x}) = M\boldsymbol{x}$, the supremum of the volume of the preimage of $C$ under $M$ when intersected with a hypersphere of radius $r$ is upper-bounded as:*

$$\sup_{\boldsymbol{c} \in \mathbb{R}^n} \text{Vol}\left[f_M^{-1}(C) \cap B_n\left[\boldsymbol{c}, r\right]\right] \le r^n \, \text{Vol}\left[\prod_{i=1}^n \left[-\frac{\beta_i}{r}, \frac{\beta_i}{r}\right] \cap B_n\left[\boldsymbol{0}, 1\right]\right], \quad (8)$$

*with*

$$\boldsymbol{\beta} = \text{abs}\left[\frac{1}{2}\Sigma^+ U^\top\right] \mathbf{1}_n + \left[\boldsymbol{0}_{\text{rank}[M]}^\top, r\mathbf{1}_{n-\text{rank}[M]}^\top\right]^\top, \quad (9)$$

*where $U\Sigma V^\top$ is the SVD decomposition of $M$ and $\Sigma^+$ is the pseudo-inverse of the diagonal matrix $\Sigma$ (i.e., the reciprocal of the non-zero elements of $\Sigma$). The volume of the box-ball intersection can be computed with Theorem 1.*

Now, Theorem 2 gives us an upper-bound for $K$, the number of images in a PSNR ball that are "collapsed" onto the same image after a linear transformation $M$.

**Bound 13: Gray image, Linear transformation (Conservative), PSNR constraint.** The capacity of a gray image $\boldsymbol{x}_g$ under a linear transformation $M \in \mathbb{R}^{cwh \times cwh}$ in the ambient space $\mathcal{A}$ is lower-bounded as

`capacity under `$M$`[in bits]` $= \log_2 n$

$$\geq \log_2 \frac{N}{K}$$

$$= \log_2 N - \log_2 K$$

$$\geq \texttt{capacity[in bits]} - cwh \log_2 \epsilon - \log_2 \text{Vol}\left(\prod_{j=1}^{cwh}\left[-\frac{\beta_j}{\epsilon}, \frac{\beta_j}{\epsilon}\right] \cap B_{cwh}\left[\mathbf{0}, 1\right]\right),$$

where $\beta$ is computed as in Theorem 2, the volume of the intersection as in Theorem 1, and `capacity[in bits]` is the capacity without the linear robustness constraint.
**Bound validity:** Assuming $cwh$ not too large (otherwise numerical evaluation of the bound becomes intractable).

This bound can be numerically unstable and is tractable only for low dimensions $cwh$, high numerical precision and large amount of sum terms kept when evaluating the intersection volume. Nevertheless, even with this extremely conservative bound, which restricts the capacity significantly more than the heuristic Bounds 10 to 12, we still observe 0.005 bpp for the most aggressive crop transform (see Table 2). This might seem small but it still amounts to over 900 bits for a $256 \times 256$px image. For the other transformations we get capacities in excess of 3,000 bits. Therefore, even in this strongly conservative setup we still see capacities significantly larger than what we observe in practice.

Bound 13 relies on a severe over-approximation of a zonotope with an axis-aligned box, meaning that it is extremely conservative. The product of singular values heuristic Bounds 10 to 12 are probably much closer to the true capacity (and in fact, possibly also conservative, as observed in the rotated ellipsoid case, Figure 8). Therefore in general we will use the heuristic Bounds 10 to 12.

### G.4 ROBUSTNESS TO COMPRESSION

Beyond geometric transformations, watermarks should also be robust to compression. Compression happens during normal image processing, even in the absence of attacks, and tends to strip a lot of information from an image. Hence, it is possible that it imposes a very strong reduction on the capacity of watermarking. The problem with compression methods, though, is that they are highly non-linear and difficult to study analytically. Here, we analyse JPEG, arguably the most widely used image compression method. While JPEG is non-linear, it has only one non-linear step. We can linearize this step without deviating too much from the behaviour of classic JPEG, resulting in LinJPEG.

The standard JPEG compression and decompression consists of the following steps (Wallace, 1991; Wikipedia, 2025):

1. Convert the colour space from `RGB` to `YCbCr`. Linear and invertible. Appendix H.2.2.

2. Downsample the `Cb` and `Cr` channels, typically by a factor of 2 in both dimensions. Linear but rank-deficient. Appendix H.2.5.

3. Divide each channel into $8 \times 8$px tiles and apply steps 4–6 to each tile individually. The `Y` channel would have $4 \times$ the tiles of the chroma channels because of the downsampling. Appendix H.2.9.

4. Perform Discrete Cosine Transform (DCT) over each tile Linear and invertible. Appendix H.2.7.

5. Do element-wise multiplication with a quantization matrix (matrix values depend on the quality setting). Linear and invertible.

6. **Round the pixel values to integers**. [non-linear, due to the rounding]

7. Perform Inverse DCT over each tile. Linear and invertible. Appendix H.2.8.

8. Upsample the `Cb` and `Cr` channels. Linear and invertible. Appendix H.2.6.

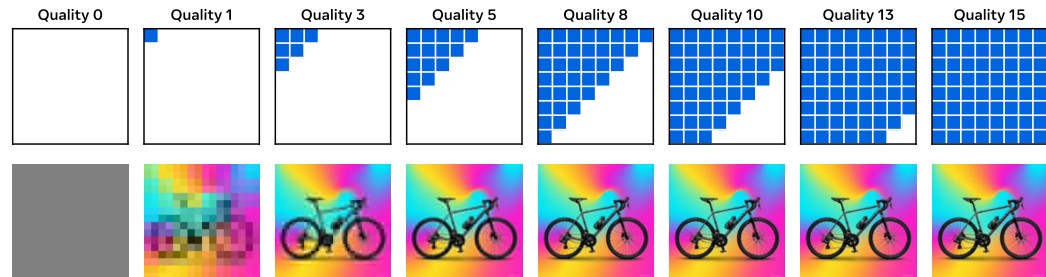

Figure 12: **Illustration of LinJPEG at various strengths.** We show the effect of compressing an image with LinJPEG, our linearised variant of JPEG compression, at various quality settings. The quality refers to the number of DCT diagonals kept for each $8{\times}8$px tile. Quality 0 means no DCT coefficients are kept, while quality 15 means that the image is unchanged. Quality 10 and above produces little visual artifacts and is almost indistinguishable from the original image.

9. Convert the color space from `YCbCr` to `RGB`. Linear and invertible. Appendix H.2.3.

The compression comes from step 2, which downsamples the chroma channels, and step 6 which efficiently encodes the frequencies which have been rounded down to 0 via entropy coding. The lossless compression after the quantization does not change the pixel values and does not affect capacity, hence we ignore it. Step 6 is the only non-linear step which prevents applying Bounds 10 to 13 to JPEG compression. Let's take a closer look at a single tile (we take the example from (Wikipedia, 2025)). When converted to DCT space ($G$), the higher frequencies are represented in the bottom and right sides of the matrix. As high-frequency components are less perceptible, the quantization matrices $Q$ are designed to attenuate them more, resulting in more of them becoming 0 after rounding:

$$G = \begin{bmatrix} -415.38 & -30.19 & -61.20 & 27.24 & 56.12 & -20.10 & -2.39 & 0.46 \\ 4.47 & -21.86 & -60.76 & 10.25 & 13.15 & -7.09 & -8.54 & 4.88 \\ -46.83 & 7.37 & 77.13 & -24.56 & -28.91 & 9.93 & 5.42 & -5.65 \\ -48.53 & 12.07 & 34.10 & -14.76 & -10.24 & 6.30 & 1.83 & 1.95 \\ 12.12 & -6.55 & -13.20 & -3.95 & -1.87 & 1.75 & -2.79 & 3.14 \\ -7.73 & 2.91 & 2.38 & -5.94 & -2.38 & 0.94 & 4.30 & 1.85 \\ -1.03 & 0.18 & 0.42 & -2.42 & -0.88 & -3.02 & 4.12 & -0.66 \\ -0.17 & 0.14 & -1.07 & -4.19 & -1.17 & -0.10 & 0.50 & 1.68 \end{bmatrix}$$

$$Q = \begin{bmatrix} 16 & 11 & 10 & 16 & 24 & 40 & 51 & 61 \\ 12 & 12 & 14 & 19 & 26 & 58 & 60 & 55 \\ 14 & 13 & 16 & 24 & 40 & 57 & 69 & 56 \\ 14 & 17 & 22 & 29 & 51 & 87 & 80 & 62 \\ 18 & 22 & 37 & 56 & 68 & 109 & 103 & 77 \\ 24 & 35 & 55 & 64 & 81 & 104 & 113 & 92 \\ 49 & 64 & 78 & 87 & 103 & 121 & 120 & 101 \\ 72 & 92 & 95 & 98 & 112 & 100 & 103 & 99 \end{bmatrix}$$

$$Q \, \mathrm{round}(G/Q) = \begin{bmatrix} -416 & -33 & -60 & 32 & 48 & -40 & 0 & 0 \\ 0 & -24 & -56 & 19 & 26 & 0 & 0 & 0 \\ -42 & 13 & 80 & -24 & -40 & 0 & 0 & 0 \\ -42 & 17 & 44 & -29 & 0 & 0 & 0 & 0 \\ 18 & 0 & 0 & 0 & 0 & 0 & 0 & 0 \\ 0 & 0 & 0 & 0 & 0 & 0 & 0 & 0 \\ 0 & 0 & 0 & 0 & 0 & 0 & 0 & 0 \\ 0 & 0 & 0 & 0 & 0 & 0 & 0 & 0 \end{bmatrix}$$

Therefore, the compression mechanism of Step 6 acts by producing zeros in the lower right corner of the DCT components $G$. That gives us a simple way to linearize Step 6: simply drop the highest frequencies, regardless of the value of their components. This is equivalent to a linear operator with a diagonal matrix with 1s and 0s on the diagonal and is explicitly constructed in Appendix H.2.10. As all the other steps are linear, we can compose them with our alternative to Step 6 to obtain LinJPEG: a linear operator that is a very close approximation to JPEG. LinJPEG is formally constructed in Appendix H.2.12. This is a similar strategy to JPEG-Mask proposed by (Zhu et al., 2018) as a way to

make JPEG differentiable. However, rather than differentiating through the compression, here we are interested in its singular values.

The way we have implemented this is to map a quality factor that is an integer from 0 to 15 inclusive to the number of diagonals that are being zeroed out. See Figure 12 for examples. While classic JPEG is designed to have fixed perceptual quality and variable file size, our LinJPEG instead has variable perceptual quality and fixed file size.

As can be seen from the capacity plot Figure 4 right for LinJPEG, the higher the compression rate, the lower the capacity, as expected. At first, it may seem strange that quality 15, where no frequency components are dropped, still has a 50% reduction in capacity. However, that is due to the downsampling of the chroma channels, which reduces the effective number of pixels we have for watermarking by half. This can also be seen by half of the singular values being 0 (Figure 11 right). Interestingly, again, even with relatively high compression rates (quality 8) we still observe capacities of more than $1$ bpp at $40$ dB. Finally, as can be seen in Table 2, even the conservative bound from Appendix G.3 gives us $0.13$ bpp or $25{,}559$ bits for a $256{\times}256$px image. Therefore, although (Lin)JPEG compression removes a lot of information from the image, it still leaves plenty of watermarking capacity.

## H  AUGMENTATIONS AS LINEAR TRANSFORMATIONS

This section describes a series of linear transformations used for the robustness bounds in Section 2.5. Each transformation is defined by a matrix $A$ and a bias vector $\boldsymbol{b}$, and acts on an input vector $\boldsymbol{x}$ as $f(\boldsymbol{x}) = A\boldsymbol{x} + \boldsymbol{b}$. Some transformations are compositions of others, as described below. We will use the pipe operator | for left-to-right composition, rather than the classical ∘ operator for right-to-left composition.

### H.1  GENERIC LINEAR TRANSFORMATION

A generic linear transformation has the form $f(\boldsymbol{x}) = A\boldsymbol{x} + \boldsymbol{b}$, where $A$ is a matrix and $\boldsymbol{b}$ is a bias vector. This transformation can be applied to vectors or images (flattened as vectors, with each third corresponding to one channel).

$$f(\boldsymbol{x}) = A\boldsymbol{x} + \boldsymbol{b}$$

### H.2  LinJPEG AND ITS BUILDING BLOCKS

#### H.2.1  PIXEL-WISE TRANSFORMATION

The pixel-wise transform applies a given linear transformation $g(\boldsymbol{p}) = B\boldsymbol{p} + \boldsymbol{c}$ independently to each pixel across an image. If the base transformation maps $c_{\text{in}}$ input channels to $c_{\text{out}}$ output channels (i.e., $B \in \mathbb{R}^{c_{\text{out}} \times c_{\text{in}}}$) the overall matrix $A$ is constructed as a block matrix, where each block is an identity matrix of size $w \times h$ (the number of pixels), scaled by the corresponding entry in the base transformation's matrix. The bias $\boldsymbol{b}$ is the base bias $\boldsymbol{c}$ repeated for every pixel.

$$A = \begin{bmatrix} B_{1,1}I_{wh} & \dots & B_{1,c_{\text{in}}}I_{wh} \\ \vdots & \ddots & \vdots \\ B_{c_{\text{out}},1}I_{wh} & \dots & B_{c_{\text{out}},c_{\text{in}}}I_{wh} \end{bmatrix} \qquad \boldsymbol{b} = \left.\begin{bmatrix} \boldsymbol{c} \\ \vdots \\ \boldsymbol{c} \end{bmatrix}\right\} wh \text{ times}$$

#### H.2.2  RGB TO YCBCR CONVERSION

Color-space transformations are pixel-wise transformations. To convert the RGB color space to the YCbCr color space we have the base matrix:

$$B = \begin{bmatrix} 0.299 & 0.587 & 0.114 \\ -0.168736 & -0.331364 & 0.5 \\ 0.5 & -0.418688 & -0.081312 \end{bmatrix}$$

The bias is $\boldsymbol{c} = [0, 128, 128]$. We can use the pixel-wise transform we defined above to obtain the linear transform for applying this across the image:

$$(A, \boldsymbol{b}) = \texttt{Pixel-wise Transformation}[B, \boldsymbol{c}].$$

#### H.2.3  YCBCR TO RGB CONVERSION

To convert the YCbCr color space back to RGB, we need the composition of two transforms: First, subtract 128 from Cb and Cr channels, then apply the matrix:

$$B = \begin{bmatrix} 1 & 0 & 1.402 \\ 1 & -0.344136 & -0.714136 \\ 1 & 1.772 & 0 \end{bmatrix}$$

Thus, we have the bias

$$\boldsymbol{c} = B[0, -128, -128]^{\top}.$$

To get a transformation for the whole image, we again use the pixel-wise transform we defined above:

$$(A, \boldsymbol{b}) = \texttt{Pixel-wise Transformation}[B, \boldsymbol{c}].$$

### H.2.4 TAKE ROWS AND COLUMNS TRANSFORMATION

This transformation selects specific rows and columns from an input image, effectively downsampling or upsampling. The matrix $A$ is constructed such that each output pixel corresponds to a specific input pixel, with a 1 at the appropriate position and 0 elsewhere. The bias $b$ is zero. Assume the indices are provided as two lists row indices and col indices. Then we can set the non-zero elements of $A$ as:

```
for ir, r in enumerate(row indices):
    for ic, c in enumerate(col indices):
        for chan in range(channels):
            A[
                chan * (len(row indices) * len(col indices)) + ir * len(col indices) + ic,
                chan * w * h + r * w + c,
            ] = 1
```

### H.2.5 DOWNSAMPLING TRANSFORMATION

The downsample transformation reduces the resolution of an image by a fixed factor. The matrix $A$ selects every $k$-th row and column. The bias $b$ is zero.

$$(A, b) = \texttt{TakeRowsAndColumns}\big[\texttt{row\_indices} = [1, 1+k, \ldots, h-k+1], \texttt{col\_indices} = [1, 1+k, \ldots, w-k+1]\big].$$

### H.2.6 UPSAMPLING TRANSFORMATION

The upsampling transformation increases the resolution by repeating rows and columns $k$ times. The matrix $A$ selects every row and column $k$ times. The bias $b$ is zero.

$$(A, b) = \texttt{TakeRowsAndColumns}\big[\texttt{row\_indices}=[\underbrace{1,\ldots,1}_{\times k},\ldots,\underbrace{h\ldots,h}_{\times k}], \texttt{col\_indices}=[\underbrace{1,\ldots,1}_{\times k},\ldots,\underbrace{w\ldots,w}_{\times k}]\big].$$

### H.2.7 DISCRETE COSINE TRANSFORM (DCT) FOR 8X8 BLOCKS

The Dct8x8 transformation applies the DCT to a single-channel $8\times8$px block. It first subtracts 128 from each pixel so that the values are centred at 0, then applies the DCT matrix. The subtraction can be done with:

$$g(\boldsymbol{x}) = I_{64}\boldsymbol{x} - 128 \cdot \mathbf{1}.$$

The DCT matrix is constructed using the four-dimensional tensor $D \in \mathbb{R}^{8\times8\times8\times8}$:

$$D_{x,y,u,v} = \cos\left(\frac{(2x+1)u\pi}{16}\right)\cos\left(\frac{(2y+1)v\pi}{16}\right)$$

$$D' = \text{reshape}(0.25\,\boldsymbol{\alpha}\boldsymbol{\alpha}^\top, (1, 64))\ \odot\ \text{reshape}(D, (64, 64)),$$

where $\boldsymbol{\alpha}_1 = 1/\sqrt{2}$, $\boldsymbol{\alpha}_i = 1$ for $2 \leq i \leq 8$ and $\odot$ is an element-wise multiplication with broadcasting. This results in

$$h(\boldsymbol{x}) = D'\boldsymbol{x}.$$

The overall transformation is:

$$f(\boldsymbol{x}) = (g\ \mid\ h)(\boldsymbol{x}).$$

### H.2.8 INVERSE DCT FOR 8X8 BLOCKS

The iDct8x8 transformation applies the inverse DCT to a single-channel $8\times8$px block. The matrix is constructed similarly to the DCT, but with the roles of $x, y$ and $u, v$ swapped. After the transformation, 128 is added to each pixel to restore the original range.

$$g(\boldsymbol{x}) = I_{64}\boldsymbol{x} + 128 \cdot \mathbf{1},$$

$$D_{x,y,u,v} = \cos\left(\frac{(2u+1)x\pi}{16}\right)\cos\left(\frac{(2v+1)u\pi}{16}\right),$$

$$D' = \text{reshape}(0.25\,\boldsymbol{\alpha}\boldsymbol{\alpha}^\top, (1, 64))\ \odot\ \text{reshape}(D, (64, 64)),$$

$$h(\boldsymbol{x}) = D'\boldsymbol{x},$$

$$f(\boldsymbol{x}) = (h\ \mid\ g)(\boldsymbol{x}).$$

### H.2.9 Tiling transformation

The tiling transformation divides an image into tiles and applies a linear transform to each tile. This operates over a single channel. Define the linear transformation for a tile as $g(\boldsymbol{t}) = B\boldsymbol{t} + \boldsymbol{c}$, with $B \in \mathbb{R}^{\mathtt{ts}^2 \times \mathtt{ts}^2}, \boldsymbol{t}, \boldsymbol{c} \in \mathbb{R}^{\mathtt{ts}^2}$, $\mathtt{ts}$ being the size of the tile (typically 8 for our use-cases). We expect that the $\mathtt{ts}$ divides the width $w$ and height $h$ of the image. The matrix $A$ is constructed by placing the base transform's matrix along the diagonal for each tile, so that each tile is transformed independently. The bias $\boldsymbol{b}$ is constructed by tiling the base bias for each tile. The $A$ and $\boldsymbol{b}$ of the resulting transformation can then be computed using this pseudo-code:

```
hor‗tiles = width // ts
ver‗tiles = height // ts

tiles = split(B, ts, axis=0)
tiles = [split(s, ts, axis=1) for s in tiles]  # ts lists of ts tiles of ts x ts size

per‗n‗rows = block‗matrix(
    [[block‗diag([tile] * hor‗tiles) for tile in htiles] for htiles in tiles]
)

A = block‗diag([per‗n‗rows] * ver‗tiles)

bias‗tiles = split(transform.bias, tile‗side, axis=0)
bias = concatenate([tile(bias‗tile, hor‗tiles) for bias‗tile in bias‗tiles])
b = tile(bias, ver‗tiles)
```

### H.2.10 JPEG filter

This filter transformation drops the lowest frequencies in an $8 \times 8$ block according to a quality factor. This is the linearized replacement for the quantization operation in the standard JPEG compression algorithm, as discussed in Appendix G.4. The quality factor $q$ designates the number of diagonals kept: from $q = 0$ for the worst quality where all diagonals are masked off, to $q = 15$ where all diagonals are kept. The resulting $A \in \mathbb{R}^{64 \times 64}$ can be constructed with the following pseudo-code:

```
matrix = triu(ones(8, 8), k=16 - 8 - q)[:, ::-1]
A = diag(matrix.flatten())
```

### H.2.11 Per-Channel transformation

The Per-Channel Transform applies a separate linear transformation to each channel of an image. The matrix $A$ is block-diagonal, with each block being the matrix for the corresponding channel. The bias $\boldsymbol{b}$ is the concatenation of the biases for each channel. Given $c$ channels, each with its own linear transformation $A_i, \boldsymbol{b}_i, 1 \leq i \leq c$, the resulting linear transform has:

$$A = \mathrm{diag}(A_1, \ldots, A_c),$$

$$\boldsymbol{b} = \begin{bmatrix} \boldsymbol{b}_1 \\ \vdots \\ \boldsymbol{b}_c \end{bmatrix}.$$

### H.2.12 LinJPEG transform

The LinJPEG transformation approximates JPEG compression with quality $q$ as a sequence of linear transforms, as explained in Appendix G.4. In a nutshell, we need to convert the colour space from RGB to YCbCr and then downsample the Cb and Cr channels by a factor of 2. We then need to apply the DCT, filter and iDCT operations on $8 \times 8$px tiles of each channel. Then we need to upsample the Cb and Cr channels to restore them to the original resolution and convert the colour space back to RGB. We can build a single linear transform with its $A$ and $\boldsymbol{b}$ doing all that by composing the building blocks defined above:

$$(A_Y, \boldsymbol{b}_Y) = \text{Tile}\left[\text{Dct8x8} \mid \text{JpegFilter}[q] \mid \text{iDct8x8}, \text{width} = w, \text{height} = h\right]$$

$$(A_C, \boldsymbol{b}_C) = \text{Downsample}[k = 2]$$

$$\mid \text{Tile}\left[\text{Dct8x8} \mid \text{JpegFilter}[q] \mid \text{iDct8x8}, \text{width} = w/2, \text{height} = h/2\right]$$

$$\mid \text{Upsample}[k = 2]$$

$$(A, \boldsymbol{b}) = \text{RGBToYCbCr} \mid \text{PerChannelTransform}[(A_Y, \boldsymbol{b}_Y), (A_C, \boldsymbol{b}_C), (A_C, \boldsymbol{b}_C)] \mid \text{YCbCrToRGB}.$$

## H.3 PIXEL MAPPING TRANSFORMS

A number of transforms can be defined via a map $\mu(x, y) \mapsto (u, v)$ that says which point $(u, v)$ in the original image corresponds to a pixel $(x, y)$ in the transformed image. Note while $x$ and $y$ are integer coordinates, $u$ and $v$ need not be. Thus, some sort of interpolation will be needed. Here we consider both nearest neighbour and bilinear interpolation. It is interesting that in both of these case, given a fixed map $\mu$, its application with interpolation is a linear operation. If $\mu$ is itself parameterized (e.g., the angle of rotation), then the transformation generally is not linear in the parameter. That is why we consider $\mu$ with different parameters to be distinct transformations.

### H.3.1 NEAREST NEIGHBOUR PIXEL MAPPING

The matrix $A$ is binary, with $A_{i,j} = 1$ if output pixel $i$ maps to input pixel $j$. Below we show the construction for a pixel map mu and a single channel. For multi-channel images, the same transformation can be applied to each channel using PerChannelTransform.

```
mesh˙x, mesh˙y = meshgrid(range(width), range(height))
mesh˙x = flatten(mesh˙x)
mesh˙y = flatten(mesh˙y)
mapped˙x, mapped˙y = vectorize(mu)(mesh˙x, mesh˙y)
mapped˙x = clip(round(mapped˙x), 0, width - 1)
mapped˙y = clip(round(mapped˙y), 0, height - 1)

A = zeros(width * height, width * height)
target˙indices = mesh˙x + mesh˙y * width
source˙indices = mapped˙x + mapped˙y * width
A[target˙indices, source˙indices] = 1
```

### H.3.2 BILINEAR PIXEL MAPPING

The matrix $A$ is constructed so that each output pixel is a weighted sum of the four nearest input pixels, with weights determined by the mapping. Below we show the construction for a pixel map mu and a single channel. For multi-channel images, the same transformation can be applied to each channel using PerChannelTransform.

```
mesh˙x, mesh˙y = meshgrid(range(width), range(height))
mesh˙x = flatten(mesh˙x)
mesh˙y = flatten(mesh˙y)
mapped˙x, mapped˙y = vectorize(mu)(mesh˙x, mesh˙y)

def get˙corners(mapped: array, range: int) -¿ tuple[array, array]:
    l = floor(mapped)
    u = ceil(mapped)
    # we need to ensure that the lower and the upper bounds are not the same
    u = where(u == l, l + 1, u)
    return l, u

mapped˙x˙l, mapped˙x˙u = get˙corners(mapped˙x, width)
mapped˙y˙l, mapped˙y˙u = get˙corners(mapped˙y, height)

denom = (mapped˙x˙u - mapped˙x˙l) * (mapped˙y˙u - mapped˙y˙l)
w11 = (mapped˙x˙u - mapped˙x) * (mapped˙y˙u - mapped˙y) / denom
w12 = (mapped˙x˙u - mapped˙x) * (mapped˙y - mapped˙y˙l) / denom
w21 = (mapped˙x - mapped˙x˙l) * (mapped˙y˙u - mapped˙y) / denom
w22 = (mapped˙x - mapped˙x˙l) * (mapped˙y - mapped˙y˙l) / denom
```

```
# Create the matrix for a single channel
target·indices = mesh·x + mesh·y * width

mapped·x·l = clip(mapped·x·l, 0, width - 1)
mapped·x·u = clip(mapped·x·u, 0, width - 1)
mapped·y·l = clip(mapped·y·l, 0, height - 1)
mapped·y·u = clip(mapped·y·u, 0, height - 1)

indices·11 = mapped·x·l + mapped·y·l * width
indices·12 = mapped·x·l + mapped·y·u * width
indices·21 = mapped·x·u + mapped·y·l * width
indices·22 = mapped·x·u + mapped·y·u * width

A = zeros(width * height, width * height)
A[target·indices, indices·11] += w11
A[target·indices, indices·12] += w12
A[target·indices, indices·21] += w21
A[target·indices, indices·22] += w22
```

### H.3.3 HORIZONTAL FLIP TRANSFORMATION

The pixel mapping is $\mu : (x, y) \mapsto (\texttt{width} - 1 - x, y)$. The transformation can be applied with either the nearest neighbour or the bilinear interpolation methods, by default we use nearest neighbour but bilinear should give the exact same result here.

### H.3.4 VERTICAL FLIP TRANSFORMATION

The pixel mapping is $\mu : (x, y) \mapsto (x, \texttt{height} - 1 - y)$. The transformation can be applied with either the nearest neighbour or the bilinear interpolation methods, by default we use nearest neighbour but bilinear should give the exact same result here.

### H.3.5 CENTRE CROP AND RESCALE TRANSFORMATION

This transformation crops the centre of an image to a given scale and rescales it to the original size. The pixel mapping $\mu$ for a fixed scale $s$ is:

$$(x, y) \mapsto \left( \left( x - \frac{\texttt{width}}{2} \right) \cdot s + \frac{\texttt{width}}{2}, \; \left( y - \frac{\texttt{height}}{2} \right) \cdot s + \frac{\texttt{height}}{2} \right).$$

The transformation can be applied with either the nearest neighbour or the bilinear interpolation methods, by default we use bilinear.

### H.3.6 ROTATION TRANSFORMATION

This rotation transformation rotates an image around its centre by a fixed angle $\theta$. The pixel mapping $\mu$ is:

$$(x, y) \mapsto \left( \left( x - \frac{\texttt{width}}{2} \right) \cos \theta - \left( y - \frac{\texttt{height}}{2} \right) \sin \theta + \frac{\texttt{width}}{2}, \right.$$

$$\left. \left( x - \frac{\texttt{width}}{2} \right) \sin \theta + \left( y - \frac{\texttt{height}}{2} \right) \cos \theta + \frac{\texttt{height}}{2} \right).$$

The transformation can be applied with either the nearest neighbour or the bilinear interpolation methods, by default we use bilinear.

# I PROOF OF THEOREM 2

Let $\boldsymbol{b}$ be a point in $\mathbb{R}^n$ and $C = \boldsymbol{b} + [0,1]^n$ be the hypercube with a corner at $\boldsymbol{b}$. Given $\boldsymbol{x} \in C$, the preimage of $f_M$ can be obtained using the Moore-Penrose pseudo-inverse $M^+$ of $M$:

$$f_M^{-1}(\boldsymbol{x}) = \left\{ M^+\boldsymbol{x} + [I - M^+M]\boldsymbol{w} \mid \boldsymbol{w} \in \mathbb{R}^n \right\}.$$

Hence:

$$
\begin{aligned}
f_M^{-1}(C) &= \left\{ M^+\boldsymbol{b} + M^+\boldsymbol{p} + (I - M^+M)\boldsymbol{w} \mid \boldsymbol{w} \in \mathbb{R}^n, \boldsymbol{p} \in [0,1]^n \right\} \\
&= \left\{ M^+\boldsymbol{b} \right\} \oplus M^+[0,1]^n \oplus (I - M^+M)\mathbb{R}^n \\
&= \left\{ M^+\left(\boldsymbol{b} + \frac{1}{2}\right) \right\} \oplus \frac{1}{2}M^+[-1,1]^n \oplus (I - M^+M)\mathbb{R}^n,
\end{aligned}
\tag{10}
$$

where $\oplus$ is the Minkowski sum. Take $U\Sigma V^\top = M$ to be the SVD decomposition of $M$. $U$ and $V$ are orthonormal matrices, while $\Sigma$ is a diagonal matrix with non-negative entries. We will assume that the singular values on the diagonal are sorted in descending order, hence the first $\text{rank}[M]$ values on the diagonal are non-zero and the rest are zero. The pseudo-inverse of $M$ can be conveniently expressed as $M^+ = V\Sigma^+U^\top$. We can use that the pseudo-inverse of diagonal matrices can be constructed by taking the reciprocals of the non-zero elements on the diagonal, leaving the zeros unchanged. Thus we have:

$$
\begin{aligned}
(I - M^+M)\mathbb{R}^n &= (I - V\Sigma^+U^\top U\Sigma V^\top)\mathbb{R}^n \\
&= (I - V\Sigma^+\Sigma V^\top)\mathbb{R}^n && (U^\top U = I \text{ as } U \text{ is orthonormal}) \\
&= (VV^\top - V\Sigma^+\Sigma V^\top)\mathbb{R}^n && (VV^\top = I \text{ as } V \text{ is orthonormal}) \\
&= V(I - \Sigma^+\Sigma)V^\top\mathbb{R}^n \\
&= V(I - \text{diag}[\mathbf{1}_{\text{rank}[M]}, \mathbf{0}_{n-\text{rank}[M]}])V^\top\mathbb{R}^n \\
&= V\,\text{diag}[\mathbf{0}_{\text{rank}[M]}, \mathbf{1}_{n-\text{rank}[M]}]V^\top\mathbb{R}^n \\
&= \tilde{V}\tilde{V}^\top\mathbb{R}^n && (\tilde{V} = V_{[:, \text{ rank}[M]+1:]} \in \mathbb{R}^{n\times(n-\text{rank}[M])}) \\
&= \tilde{V}\mathbb{R}^{n-\text{rank}[M]} && (\tilde{V}^\top\mathbb{R}^n = \mathbb{R}^{n-\text{rank}[M]}).
\end{aligned}
$$

Thus, combining with Equation (10), we have that the set of images that would be mapped by $M$ to images in the cube $C$, i.e., after quantization, is the following polytope:

$$f_M^{-1}(C) = \left\{ M^+\left(\boldsymbol{b} + \frac{1}{2}\right) \right\} \oplus \frac{1}{2}V\Sigma^+U^\top[-1,1]^n \oplus \tilde{V}\mathbb{R}^{n-\text{rank}[M]}.$$

The volume of the intersection in Equation (8) is maximized when the ball centre $\boldsymbol{c}$ coincides with the polytope centre $M^+(\boldsymbol{b} + 1/2)$. Furthermore, because the volume of the intersection is invariant to shifts, we can simplify the left-hand side of Equation (8) to:

$$
\begin{aligned}
\sup_{\boldsymbol{c}\in\mathbb{R}^n} \text{Vol}\left[ f_M^{-1}(C) \cap B_n[\boldsymbol{c}, r] \right] &= \text{Vol}\left[ \left( \frac{1}{2}V\Sigma^+U^\top[-1,1]^n \oplus \tilde{V}\mathbb{R}^{n-\text{rank}[M]} \right) \cap B_n[\mathbf{0}, r] \right] \\
&= \text{Vol}\left[ \left( \underbrace{\frac{1}{2}V\Sigma^+U^\top[-1,1]^n \oplus r\tilde{V}[-1,1]^{n-\text{rank}[M]}}_{Z} \right) \cap B_n[\mathbf{0}, r] \right],
\end{aligned}
$$

where we use the fact that the intersection with a ball of radius $r$ needs to be contained within the hypercube $[-r, r]^n$, that $\tilde{V}$ has orthonormal columns and that the two sets making up the sum in $Z$ are orthogonal.[4]

Computing the volume of this intersection in the general case is computationally intractable. However, if we over-approximate the left-hand side of the intersection ($Z$) with a (rotated) box, then we can use our previous results on the volume of box-ball intersections, in particular Theorem 1.

---

[4]Follows from the properties of the pseudo-inverse: $\left(M^+M\right)^\top = M^+M$ and $M^+MM^+ = M^+$.

To upper-bound $Z$ with a (rotated) box we first observe that it is the Minkowski sum of two zonotopes and hence is a zonotope itself. A zonotope is defined as

$$\left\{ \boldsymbol{x} \in \mathbb{R}^n : \boldsymbol{x} = \boldsymbol{c} + \sum_{i=1}^{p} \xi_i \boldsymbol{g}_i, \ \xi_i \in [-1,1] \ \forall i = 1, \dots, p \right\},$$

where $\boldsymbol{c} \in \mathbb{R}^n$ is its centre and $G = \{\boldsymbol{g}_i\}_{i=1}^{p}$, $\boldsymbol{g}_i \in \mathbb{R}^n$ is the set of its generators. A zonotope is, equivalently the Minkowski sum of line segments. Thus, the Minkowski sum of zonotopes is also a zonotope. Its centre is found by adding together the centres of the original zonotopes, and its set of generators is just all the generators from both shapes combined (Schneider, 2013). Now, it is clear that $Z$ is a zonotope as it is the Minkowski sum of two other zonotopes.

The $V$ and $\tilde{V}$ matrices simply rotate the resulting zonotope. As the ball $B_n\left[\boldsymbol{0}, r\right]$ is rotation invariant, we can ignore this rotation. That will help us with tightening the box approximation as the $n - \text{rank}[M]$ dimensions of the second zonotope will now be automatically axis-aligned. Thus we have (up to a rotation):

$$Z = \left[\tfrac{1}{2}\Sigma^+ U^\top, \quad r I_{n[:,\text{rank } M+1:]}\right] [-1,1]^{2n - \text{rank } M} = G\left[-1,1\right]^{2n - \text{rank } M},$$

with $G \in \mathbb{R}^{n \times (2n - \text{rank } M)}$ being its $2n - \text{rank } M$ $n$-dimensional generators.

A zonotope with generators $G$ is contained in an axis-aligned box $\prod_{i=1}^{n}[-\beta_i', \beta_i']$, where $\boldsymbol{\beta}$ is the sum of absolute values of $G$ across the generators: $\boldsymbol{\beta}' = \text{abs}[G] \, \mathbf{1}_{2n - \text{rank}[M]} = \text{abs}[\tfrac{1}{2}\Sigma^+ U^\top]\mathbf{1}_n + [\mathbf{0}_{\text{rank}[M]}^\top, r\mathbf{1}_{n-\text{rank}[M]}^\top]^\top$ (Girard, 2005; Althoff et al., 2010).

Note that this over-approximation can be extremely loose: this is the step that makes Bound 13 so conservative. However, with this, we can now apply Equation (4) and Theorem 1 for the box-ball intersection:

$$
\begin{aligned}
\sup_{\boldsymbol{c} \in \mathbb{R}^n} \text{Vol}\left[f_M^{-1}(C) \cap B_n\left[\boldsymbol{c}, r\right]\right] &= \text{Vol}\left[\left(\frac{1}{2}V\Sigma^+ U^\top[-1,1]^n \oplus r\tilde{V}[-1,1]^{n-\text{rank}[M]}\right) \cap B_n\left[\boldsymbol{0}, r\right]\right] \\
&= \text{Vol}\left[\left(G\left[-1,1\right]^{2n-\text{rank}[M]}\right) \cap B_n\left[\boldsymbol{0}, r\right]\right] \\
&\leq \text{Vol}\left[([-\beta_1, \beta_1] \times \cdots \times [-\beta_n, \beta_n]) \cap B_n\left[\boldsymbol{0}, r\right]\right] \\
&= r^n \, \text{Vol}\left[\left(\left[-\frac{\beta_1}{r}, \frac{\beta_1}{r}\right] \times \cdots \times \left[-\frac{\beta_n}{r}, \frac{\beta_n}{r}\right]\right) \cap B_n\left[\boldsymbol{0}, 1\right]\right].
\end{aligned}
$$

# J Comprehensive results for ChunkySeal

## J.1 Extended evaluation

Table 4: **Extended results of Chunky Seal on SA-1B (Kirillov et al., 2023).**

| | Chunky Seal (ours) 1024bit, 256px | Video Seal 256bit, 256px | Video Seal 96bit, 256px | HiDDeN | MBRS | TrustMark | WAM |
|---|---|---|---|---|---|---|---|
| Capacity | 1024 bits | 256 bits | 96 bits | 48 bits | 256 bits | 100 bits | 32 bits |
| PSNR | 45.32 dB | 44.42 dB | 53.19 dB | 30.41 dB | 45.54 dB | 42.29 dB | 38.19 dB |
| SSIM | 0.9945 | 0.9963 | 0.9995 | 0.9299 | 0.9962 | 0.9941 | 0.9842 |
| MS-SSIM | 0.9966 | 0.9972 | 0.9993 | 0.9062 | 0.9967 | 0.9944 | 0.9877 |
| LPIPS | 0.0085 | 0.0019 | 0.0028 | 0.2021 | 0.0044 | 0.0028 | 0.0446 |
| Embedding Time | 0.27 s | 0.06 s | 0.06 s | 0.08 s | 0.08 s | 0.07 s | 0.15 s |
| Extraction Time | 0.05 s | 0.01 s | 0.01 s | 0.01 s | 0.01 s | 0.01 s | 0.02 s |
| Bit Acc. | 99.74% | 99.90% | 97.92% | 92.19% | 98.74% | 99.81% | 100.00% |
| Bit Acc. (Horizontal Flip) | 99.65% | 99.89% | 97.20% | 64.06% | 50.63% | 99.81% | 100.00% |
| Bit Acc. (Rotate 5°) | 99.29% | 99.37% | 94.79% | 80.99% | 50.21% | 56.88% | 98.83% |
| Bit Acc. (Rotate 10°) | 97.26% | 98.31% | 90.26% | 72.22% | 50.42% | 48.23% | 75.00% |
| Bit Acc. (Rotate 30°) | 49.56% | 51.30% | 54.93% | 50.09% | 51.50% | 49.65% | 51.82% |
| Bit Acc. (Rotate 45°) | 50.01% | 50.18% | 50.20% | 46.88% | 51.50% | 50.96% | 50.91% |
| Bit Acc. (Rotate 90°) | 51.37% | 81.07% | 49.08% | 49.57% | 50.50% | 49.42% | 50.78% |
| Bit Acc. (Resize 32%) | 99.75% | 99.89% | 97.92% | 90.36% | 98.24% | 99.81% | 100.00% |
| Bit Acc. (Resize 45%) | 99.73% | 99.90% | 97.88% | 91.06% | 98.51% | 99.81% | 100.00% |
| Bit Acc. (Resize 55%) | 99.73% | 99.90% | 97.88% | 91.23% | 98.60% | 99.81% | 100.00% |
| Bit Acc. (Resize 63%) | 99.74% | 99.90% | 97.92% | 91.67% | 98.65% | 99.81% | 100.00% |
| Bit Acc. (Resize 71%) | 99.73% | 99.90% | 97.96% | 91.75% | 98.69% | 99.81% | 100.00% |
| Bit Acc. (Resize 77%) | 99.74% | 99.89% | 97.88% | 91.75% | 98.68% | 99.81% | 100.00% |
| Bit Acc. (Resize 84%) | 99.74% | 99.90% | 97.92% | 92.01% | 98.68% | 99.81% | 100.00% |
| Bit Acc. (Resize 89%) | 99.74% | 99.90% | 97.92% | 91.84% | 98.69% | 99.81% | 100.00% |
| Bit Acc. (Resize 95%) | 99.74% | 99.90% | 97.92% | 92.01% | 98.71% | 99.81% | 100.00% |
| Bit Acc. (Crop 32%) | 49.71% | 50.24% | 50.84% | 48.70% | 49.95% | 49.65% | 79.30% |
| Bit Acc. (Crop 45%) | 65.22% | 50.70% | 51.52% | 48.35% | 50.59% | 51.73% | 94.14% |
| Bit Acc. (Crop 55%) | 86.90% | 52.73% | 63.58% | 57.03% | 50.20% | 51.58% | 96.22% |
| Bit Acc. (Crop 63%) | 93.42% | 66.70% | 79.13% | 64.06% | 49.77% | 51.81% | 97.79% |
| Bit Acc. (Crop 71%) | 95.85% | 87.34% | 86.74% | 69.44% | 50.26% | 56.42% | 98.83% |
| Bit Acc. (Crop 77%) | 97.13% | 95.33% | 91.83% | 74.39% | 50.57% | 92.85% | 99.35% |
| Bit Acc. (Crop 84%) | 97.77% | 98.31% | 93.47% | 79.86% | 50.59% | 99.88% | 99.61% |
| Bit Acc. (Crop 89%) | 98.81% | 98.97% | 94.83% | 82.47% | 50.38% | 99.92% | 99.22% |
| Bit Acc. (Crop 95%) | 99.29% | 99.56% | 95.03% | 82.90% | 50.93% | 99.92% | 99.22% |
| Bit Acc. (Brightness 10%) | 83.62% | 83.17% | 82.69% | 51.13% | 61.76% | 80.04% | 96.48% |
| Bit Acc. (Brightness 25%) | 99.57% | 98.93% | 95.43% | 56.25% | 84.39% | 97.19% | 100.00% |
| Bit Acc. (Brightness 50%) | 99.74% | 99.76% | 97.48% | 75.52% | 95.01% | 99.54% | 100.00% |
| Bit Acc. (Brightness 75%) | 99.73% | 99.84% | 98.04% | 86.81% | 97.91% | 99.62% | 100.00% |
| Bit Acc. (Brightness 125%) | 99.22% | 98.91% | 94.47% | 94.36% | 95.42% | 97.00% | 100.00% |
| Bit Acc. (Brightness 150%) | 97.26% | 96.18% | 87.78% | 93.23% | 91.03% | 92.42% | 100.00% |
| Bit Acc. (Brightness 175%) | 95.74% | 94.48% | 83.13% | 93.06% | 88.79% | 90.35% | 99.22% |
| Bit Acc. (Brightness 200%) | 95.10% | 92.85% | 80.17% | 92.88% | 86.93% | 88.31% | 99.48% |
| Bit Acc. (Contrast 10%) | 99.46% | 97.88% | 84.21% | 51.65% | 74.01% | 74.96% | 95.31% |
| Bit Acc. (Contrast 25%) | 99.67% | 99.76% | 95.91% | 56.25% | 89.35% | 95.19% | 100.00% |
| Bit Acc. (Contrast 50%) | 99.74% | 99.82% | 97.56% | 75.95% | 96.21% | 99.19% | 100.00% |
| Bit Acc. (Contrast 75%) | 99.74% | 99.85% | 97.92% | 86.89% | 98.08% | 99.54% | 100.00% |
| Bit Acc. (Contrast 125%) | 99.58% | 99.59% | 95.15% | 94.70% | 96.63% | 98.65% | 100.00% |
| Bit Acc. (Contrast 150%) | 99.11% | 98.96% | 92.35% | 96.09% | 93.69% | 95.23% | 100.00% |
| Bit Acc. (Contrast 175%) | 98.52% | 97.49% | 89.58% | 96.09% | 91.50% | 92.15% | 99.87% |
| Bit Acc. (Contrast 200%) | 97.86% | 95.88% | 86.98% | 96.61% | 89.21% | 90.08% | 99.74% |
| Bit Acc. (Hue -0.2) | 98.37% | 99.40% | 83.25% | 60.68% | 95.39% | 97.31% | 95.18% |
| Bit Acc. (Hue -0.1) | 99.66% | 99.56% | 94.63% | 73.09% | 97.28% | 98.92% | 99.87% |
| Bit Acc. (Hue 0.1) | 99.68% | 99.06% | 95.47% | 80.82% | 97.12% | 98.54% | 100.00% |
| Bit Acc. (Hue 0.2) | 99.28% | 99.04% | 81.57% | 59.46% | 95.70% | 97.50% | 98.70% |
| Bit Acc. (JPEG 40) | 97.18% | 99.41% | 94.91% | 91.32% | 98.35% | 99.50% | 100.00% |
| Bit Acc. (JPEG 50) | 98.35% | 99.64% | 96.47% | 91.58% | 98.84% | 99.62% | 100.00% |
| Bit Acc. (JPEG 60) | 98.62% | 99.76% | 96.15% | 91.49% | 98.54% | 99.62% | 100.00% |
| Bit Acc. (JPEG 70) | 98.90% | 99.74% | 96.39% | 91.49% | 98.42% | 99.62% | 100.00% |
| Bit Acc. (JPEG 80) | 99.31% | 99.84% | 97.40% | 91.84% | 98.60% | 99.69% | 100.00% |
| Bit Acc. (JPEG 90) | 99.60% | 99.85% | 97.32% | 92.10% | 98.66% | 99.69% | 100.00% |
| Bit Acc. (Gaussian Blur 3) | 99.74% | 99.90% | 97.92% | 91.67% | 98.66% | 99.81% | 100.00% |
| Bit Acc. (Gaussian Blur 5) | 99.74% | 99.90% | 97.96% | 90.97% | 98.47% | 99.81% | 100.00% |
| Bit Acc. (Gaussian Blur 9) | 99.74% | 99.89% | 97.88% | 89.24% | 98.15% | 99.81% | 100.00% |
| Bit Acc. (Gaussian Blur 13) | 99.73% | 99.85% | 97.96% | 87.15% | 97.67% | 99.77% | 100.00% |
| Bit Acc. (Gaussian Blur 17) | 99.70% | 99.79% | 98.00% | 84.98% | 96.83% | 99.73% | 100.00% |

Table 5: **Extended results of Chunky Seal on COCO (Lin et al., 2014).**

| | Chunky Seal (ours) 1024bit, 256px | Video Seal 256bit, 256px | Video Seal 96bit, 256px | HiDDeN | MBRS | TrustMark | WAM |
|---|---|---|---|---|---|---|---|
| Capacity | 1024 bits | 256 bits | 96 bits | 48 bits | 256 bits | 100 bits | 32 bits |
| PSNR | 44.29 dB | 44.94 dB | 53.33 dB | 30.51 dB | 45.81 dB | 42.72 dB | 38.73 dB |
| SSIM | 0.9917 | 0.9953 | 0.9992 | 0.8469 | 0.9944 | 0.9921 | 0.9803 |
| MS-SSIM | 0.9968 | 0.9975 | 0.9988 | 0.9203 | 0.9976 | 0.9931 | 0.9891 |
| LPIPS | 0.0061 | 0.0022 | 0.0033 | 0.1850 | 0.0035 | 0.0015 | 0.0295 |
| Embedding Time | 0.03 s | 0.01 s | 0.01 s | 0.01 s | 0.01 s | 0.01 s | 0.03 s |
| Extraction Time | 0.04 s | 0.01 s | 0.01 s | 0.00 s | 0.00 s | 0.00 s | 0.01 s |
| Bit Acc. | 99.66% | 99.92% | 97.64% | 92.40% | 98.70% | 99.90% | 100.00% |
| Bit Acc. (Horizontal Flip) | 99.52% | 99.87% | 97.16% | 61.83% | 49.87% | 99.87% | 99.97% |
| Bit Acc. (Rotate 5°) | 97.11% | 99.06% | 94.69% | 79.85% | 50.04% | 65.31% | 97.84% |
| Bit Acc. (Rotate 10°) | 94.46% | 97.56% | 91.53% | 72.65% | 49.92% | 51.89% | 77.16% |
| Bit Acc. (Rotate 30°) | 50.57% | 50.83% | 57.08% | 53.37% | 49.64% | 50.05% | 51.00% |
| Bit Acc. (Rotate 45°) | 49.98% | 50.53% | 50.55% | 49.54% | 50.15% | 50.93% | 50.75% |
| Bit Acc. (Rotate 90°) | 56.36% | 83.44% | 50.58% | 51.23% | 49.90% | 49.84% | 49.28% |
| Bit Acc. (Resize 32%) | 94.69% | 97.69% | 97.53% | 71.19% | 90.57% | 99.81% | 99.81% |
| Bit Acc. (Resize 45%) | 98.60% | 99.56% | 97.70% | 80.50% | 96.16% | 99.86% | 100.00% |
| Bit Acc. (Resize 55%) | 99.31% | 99.79% | 97.64% | 84.88% | 97.28% | 99.87% | 100.00% |
| Bit Acc. (Resize 63%) | 99.53% | 99.86% | 97.66% | 87.17% | 97.80% | 99.90% | 100.00% |
| Bit Acc. (Resize 71%) | 99.63% | 99.89% | 97.67% | 87.92% | 98.11% | 99.88% | 100.00% |
| Bit Acc. (Resize 77%) | 99.66% | 99.91% | 97.69% | 88.71% | 98.21% | 99.90% | 100.00% |
| Bit Acc. (Resize 84%) | 99.66% | 99.91% | 97.68% | 89.23% | 98.34% | 99.90% | 100.00% |
| Bit Acc. (Resize 89%) | 99.67% | 99.93% | 97.67% | 89.38% | 98.34% | 99.91% | 100.00% |
| Bit Acc. (Resize 95%) | 99.66% | 99.92% | 97.70% | 89.75% | 98.43% | 99.89% | 100.00% |
| Bit Acc. (Crop 32%) | 49.84% | 50.25% | 50.75% | 49.27% | 49.54% | 49.95% | 73.88% |
| Bit Acc. (Crop 45%) | 60.64% | 49.73% | 50.36% | 50.40% | 50.04% | 50.79% | 91.47% |
| Bit Acc. (Crop 55%) | 82.26% | 50.59% | 59.70% | 56.02% | 49.99% | 49.79% | 95.72% |
| Bit Acc. (Crop 63%) | 89.82% | 58.23% | 74.25% | 61.19% | 50.64% | 51.19% | 97.19% |
| Bit Acc. (Crop 71%) | 93.68% | 81.43% | 85.59% | 67.94% | 49.80% | 54.77% | 97.22% |
| Bit Acc. (Crop 77%) | 94.71% | 92.29% | 89.15% | 73.06% | 49.70% | 87.79% | 98.19% |
| Bit Acc. (Crop 84%) | 96.04% | 97.64% | 92.94% | 78.83% | 49.66% | 99.95% | 99.12% |
| Bit Acc. (Crop 89%) | 97.17% | 98.96% | 94.32% | 81.27% | 49.84% | 99.98% | 98.69% |
| Bit Acc. (Crop 95%) | 97.88% | 99.48% | 95.21% | 83.31% | 50.82% | 99.96% | 99.22% |
| Bit Acc. (Brightness 10%) | 85.06% | 81.38% | 81.43% | 52.33% | 62.51% | 83.43% | 96.12% |
| Bit Acc. (Brightness 25%) | 98.76% | 98.82% | 94.93% | 55.85% | 84.75% | 97.85% | 99.75% |
| Bit Acc. (Brightness 50%) | 99.53% | 99.84% | 97.06% | 74.29% | 95.43% | 99.74% | 100.00% |
| Bit Acc. (Brightness 75%) | 99.65% | 99.92% | 97.50% | 86.44% | 97.98% | 99.87% | 100.00% |
| Bit Acc. (Brightness 125%) | 99.16% | 99.02% | 96.24% | 95.04% | 95.24% | 98.56% | 100.00% |
| Bit Acc. (Brightness 150%) | 98.47% | 97.78% | 94.73% | 95.56% | 92.20% | 96.45% | 100.00% |
| Bit Acc. (Brightness 175%) | 97.34% | 96.13% | 92.64% | 95.63% | 89.16% | 94.62% | 100.00% |
| Bit Acc. (Brightness 200%) | 95.83% | 94.18% | 89.48% | 94.92% | 86.91% | 91.97% | 99.97% |
| Bit Acc. (Contrast 10%) | 97.77% | 98.29% | 83.32% | 52.31% | 77.41% | 78.54% | 93.69% |
| Bit Acc. (Contrast 25%) | 99.43% | 99.68% | 95.15% | 55.79% | 90.76% | 96.47% | 99.78% |
| Bit Acc. (Contrast 50%) | 99.62% | 99.87% | 97.11% | 73.81% | 96.71% | 99.68% | 100.00% |
| Bit Acc. (Contrast 75%) | 99.66% | 99.92% | 97.50% | 86.27% | 98.19% | 99.87% | 100.00% |
| Bit Acc. (Contrast 125%) | 99.23% | 99.19% | 95.83% | 94.67% | 95.67% | 98.57% | 100.00% |
| Bit Acc. (Contrast 150%) | 98.55% | 97.84% | 93.30% | 96.00% | 92.83% | 95.83% | 99.97% |
| Bit Acc. (Contrast 175%) | 97.75% | 96.42% | 91.00% | 96.58% | 90.38% | 93.65% | 99.97% |
| Bit Acc. (Contrast 200%) | 96.89% | 95.07% | 89.08% | 96.85% | 88.58% | 90.87% | 99.88% |
| Bit Acc. (Hue -0.2) | 97.37% | 98.35% | 82.19% | 59.62% | 95.41% | 99.03% | 96.91% |
| Bit Acc. (Hue -0.1) | 99.41% | 98.77% | 94.92% | 72.27% | 97.34% | 99.73% | 100.00% |
| Bit Acc. (Hue 0.1) | 99.37% | 99.05% | 95.53% | 82.04% | 97.63% | 99.76% | 99.97% |
| Bit Acc. (Hue 0.2) | 97.70% | 98.64% | 78.98% | 61.85% | 96.64% | 99.54% | 98.06% |
| Bit Acc. (JPEG 40) | 65.86% | 97.79% | 72.64% | 87.98% | 95.44% | 98.34% | 97.28% |
| Bit Acc. (JPEG 50) | 72.47% | 98.74% | 80.22% | 88.21% | 96.22% | 99.05% | 98.31% |
| Bit Acc. (JPEG 60) | 76.97% | 99.26% | 85.39% | 88.44% | 97.28% | 99.38% | 98.84% |
| Bit Acc. (JPEG 70) | 82.93% | 99.55% | 89.42% | 88.77% | 97.50% | 99.64% | 99.53% |
| Bit Acc. (JPEG 80) | 88.89% | 99.79% | 93.34% | 89.31% | 97.99% | 99.73% | 99.56% |
| Bit Acc. (JPEG 90) | 93.21% | 99.87% | 96.06% | 90.50% | 98.25% | 99.81% | 99.84% |
| Bit Acc. (Gaussian Blur 3) | 99.63% | 99.89% | 97.70% | 86.23% | 97.75% | 99.87% | 100.00% |
| Bit Acc. (Gaussian Blur 5) | 99.29% | 99.86% | 97.75% | 80.40% | 96.09% | 99.86% | 100.00% |
| Bit Acc. (Gaussian Blur 9) | 97.95% | 99.74% | 97.55% | 71.00% | 90.90% | 99.86% | 99.88% |
| Bit Acc. (Gaussian Blur 13) | 93.88% | 99.57% | 97.12% | 65.04% | 84.27% | 99.85% | 99.62% |
| Bit Acc. (Gaussian Blur 17) | 84.65% | 99.11% | 96.64% | 60.94% | 77.49% | 99.41% | 99.38% |

## J.2 IMAGE EXAMPLES

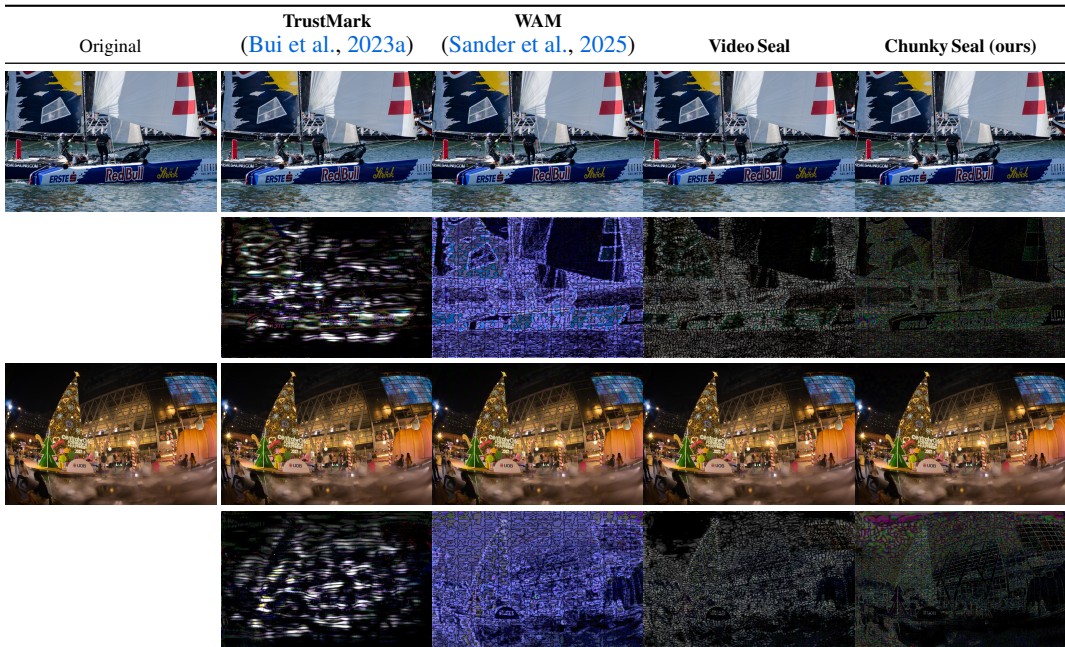

Figure 13: Qualitative results for the different watermarking methods on images taken from the SA-1b dataset at their original resolutions. We show the original images, the watermarked ones, and the watermark distortions brightened for clarity.

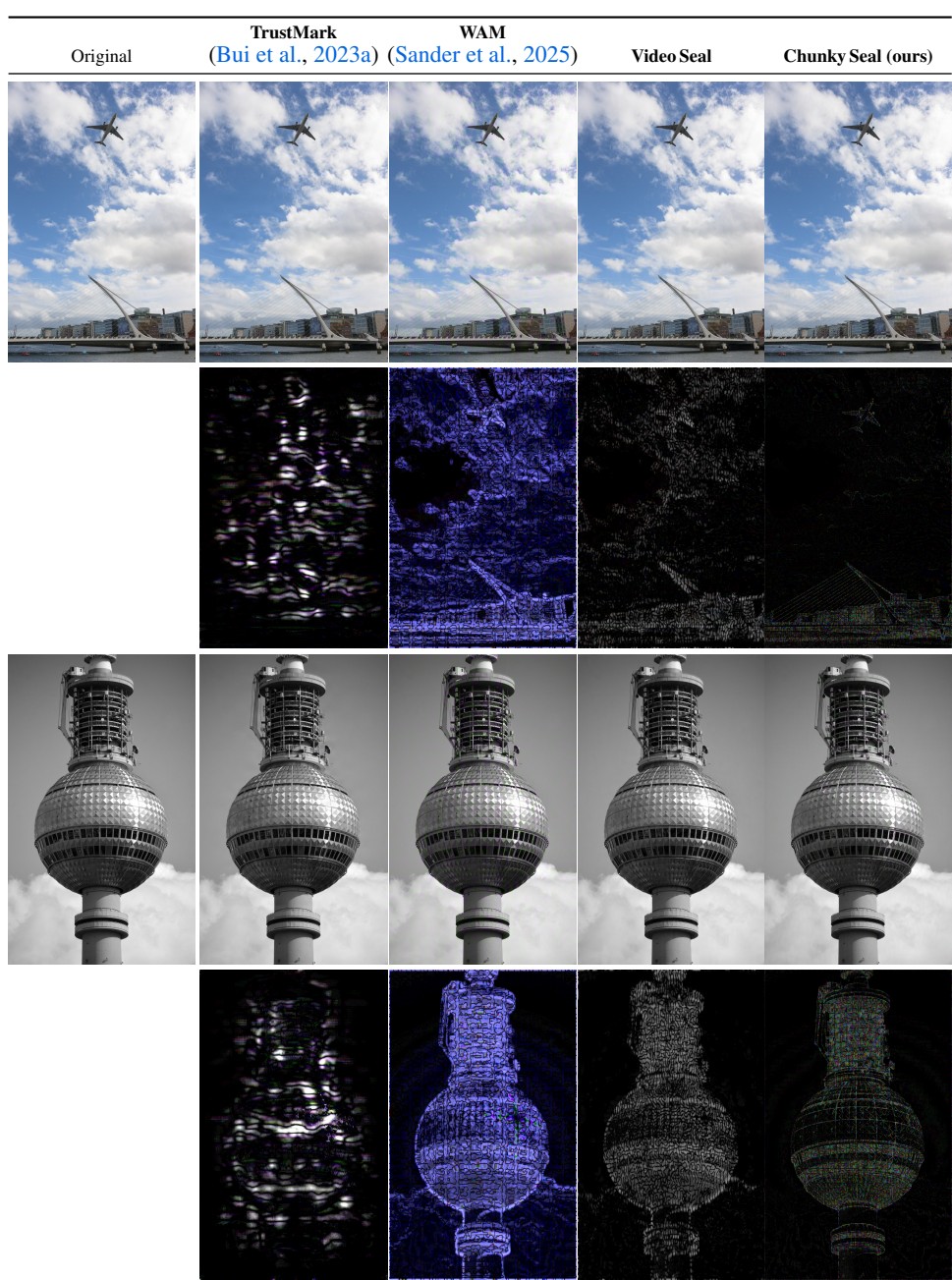

Figure 14: Qualitative results for the different watermarking methods on images taken from the SA-1b dataset at their original resolutions. We show the original images, the watermarked ones, and the watermark distortions brightened for clarity.

