# OpenReview forum: "We Can Hide More Bits: The Unused Watermarking Capacity in Theory and in Practice"
_ICLR.cc/2026/Conference — Submitted to ICLR 2026_

### Official Review · Reviewer_851W · 2025-10-25

**Soundness:** 3
**Presentation:** 1
**Contribution:** 2
**Rating:** 4
**Confidence:** 2

**Summary:**

This paper investigates the theoretical capacity of image watermarking and highlights the large gap between theoretical upper bounds and practical deep learning performance. The authors derive capacity bounds under PSNR constraints by interpreting PSNR as an equivalent l2-ball constraint within the image cube. By counting integer points within the cube-ball intersection, they estimate the achievable bit capacity and demonstrate that current models (e.g., VideoSeal) use only a fraction of the potential.
Empirical results show that under a simple grayscale + PSNR setup, VideoSeal fails to reliably encode 1024 bits, while linear or handcrafted embedding-decoding schemes can succeed with 1024–2048 bits. The proposed method, ChunkySeal,  demonstrate that scaling up the baseline achieves 4× higher capacity (256 → 1024 bits) while maintaining quality and robustness, but still remains far from the theoretical limits.

**Strengths:**

- Clear theoretical framework linking PSNR and l2 constraints : The cube–sphere intersection formulation provides intuitive and quantitative insight into capacity bounds.
- Well-controlled experiments isolating key variables : Simplified settings (single grayscale image, PSNR constraint only) help pinpoint that architectural/optimization limits (not data or format) cause the current capacity gap.
- Practical contribution (ChunkySeal) : A straightforward scaling of the embedder/extractor boosts performance, serving as a sanity check and strong baseline for future work.
- Inclusion of robustness considerations: The paper extends its theory to cover linearized transformations (LinJPEG, rotation, scaling), proposing heuristic and conservative bounds.

**Weaknesses:**

- Limited formal robustness analysis : The provided bounds for transformations (Bounds 10–13) are heuristic or overly conservative. Non-linear effects such as quantization and rounding are not analytically handled.
- PSNR as a potentially weak perceptual proxy : The paper’s reliance on PSNR ignores perceptual discrepancies—two images with identical PSNR can differ visually. Extensions using LPIPS or MS-SSIM would better reflect real-world perceptual constraints.
- Lack of in-depth analysis on model failure causes : While VideoSeal’s underperformance is empirically demonstrated, the architectural or optimization bottlenecks (e.g., skip connections, normalization, bandwidth limits) are not deeply dissected.
- Simplified image-space assumption : The capacity derivation assumes a BMP-like uncompressed pixel grid. Real-world formats (JPEG, PNG) involve non-linear compression steps not fully captured, even with the LinJPEG approximation.

**Questions:**

- Validity of PSNR and L2-ball equivalence : Have you tested whether two perturbations with equal PSNR but different visual artifacts yield consistent capacity results? Would using perceptual metrics (LPIPS, MS-SSIM) alter the theoretical limit?
- Failure analysis of VideoSeal : What specifically prevents VideoSeal from scaling beyond 1024 bits—optimization instability, insufficient representation capacity, or architectural bottlenecks? Any diagnostic results (e.g., layer-wise activation spectra) to support this?
- Completeness of Figure 2 cases : Does Figure 2 fully capture all geometric cases? What happens if the sphere’s center lies along cube edges or planes (partial overlap)? Are there discontinuities or nonlinear capacity changes in these intermediate configurations?
- Image format generalization : How would your capacity estimation adapt to real formats like JPEG (non-linear quantization) or PNG (filter-based compression)? Can LinJPEG capture these effects accurately, or are there measurable deviations?
- Theoretical limits vs. hyperparameter tuning : If a theoretical limit exists, why can’t it be reached through simple hyperparameter sweeps (e.g., reconstruction loss weight )? Is the gap due to optimization dynamics or representational constraints? A quantitative analysis (e.g., singular value decomposition of the embedding mapping) would clarify this.

Efficiency and practicality of ChunkySeal.ChunkySeal reaches higher bit capacity but at the cost of ~760M parameters. How feasible is this in deployment scenarios? Could lighter architectures (e.g., tiled embeddings, structured transforms) achieve similar performance?

---

> ### Author Response · Authors · 2025-11-21
>
> We are happy to hear that you find our theoretical framework clear and intuitive, our experiments well-controlled and helpful, and our proposed ChunkySeal to be a strong baseline for future work. This is precisely what we were aiming at. We would like to nevertheless address your concerns and questions.
>
> > Limited formal robustness analysis : The provided bounds for transformations (Bounds 10–13) are heuristic or overly conservative. Non-linear effects such as quantization and rounding are not analytically handled.
>
> While Bounds 10–12 do have heuristic elements and Bound 13 is overly conservative, they work together to complement each other. Furthermore, as shown in Table 2 in the manuscript, even the overly conservative Bound 13 allows for thousands of bits to be embedded for most setups. **Thus, being conservative does not diminish its utility in any way whatsoever.**
>
> **Furthermore, it is not correct that “non-linear effects such as quantization and rounding are not analytically handled”.** The conservative Bound 13 (via Theorem 2) handles these effects by upper-bounding the preimage of the box of all continuous-valued images that would get rounded to the same quantized image.
>
> > PSNR as a potentially weak perceptual proxy: The paper’s reliance on PSNR ignores perceptual discrepancies—two images with identical PSNR can differ visually. Extensions using LPIPS or MS-SSIM would better reflect real-world perceptual constraints.
>
> We agree that PSNR is a weak perceptual proxy. However, it is precisely its simplicity that allows the theoretical analysis in the paper. For this reason, **we do not foresee it being feasible to do a similar theoretical analysis using LPIPS or MS-SSIM**. That being said, our evaluation of ChunkySeal do include LPIPS or MS-SSIM where we observe comparable quality to VideoSeal.
>
> > Validity of PSNR and L2-ball equivalence : Have you tested whether two perturbations with equal PSNR but different visual artifacts yield consistent capacity results?
>
> Our capacity results depend only on the cover image —whether it is gray (for the highest possible capacity) or corner image (for the lowest)— and the maximum allowable PSNR. They do not depend on the choice of watermarked images. Thus, evaluating our results for the same mode of cover (gray/corner) at the same PSNR results in the same capacity.
>
> > Simplified image-space assumption : The capacity derivation assumes a BMP-like uncompressed pixel grid. Real-world formats (JPEG, PNG) involve non-linear compression steps not fully captured, even with the LinJPEG approximation.
>
> > Image format generalization : How would your capacity estimation adapt to real formats like JPEG (non-linear quantization) or PNG (filter-based compression)? Can LinJPEG capture these effects accurately, or are there measurable deviations?
>
> **PNG uses lossless compression and hence has no impact on the capacity of the watermark.** LinJPEG fully captures every single step of the JPEG compression. The only non-linear step in JPEG (the choice of which frequency components are dropped) is replaced with a fixed choice (given a quality setting) for this set of components. However, this has little bearing to the amount of information being destroyed by the compression. **For the purposes of the theoretical analysis, JPEG and LinJPEG can be considered to be essentially the same.**
>
> > Completeness of Figure 2 cases : Does Figure 2 fully capture all geometric cases? What happens if the sphere’s center lies along cube edges or planes (partial overlap)? Are there discontinuities or nonlinear capacity changes in these intermediate configurations?
>
> Great question! We considered the best (center, gray image) and the worst (corner, fully saturated image) scenarios. All other images would fall in between. In the case when the cover image lies along the edge (or an $n$-face) of the cube would correspond to an image that has ($cwh - n$) of its pixels saturated and $n$ pixels being not saturated. This would have strictly higher capacity than the fully saturated corner case. The relationship between the location of the cover image within the cube and the capacity is nonlinear but continuous. The capacity is also monotonically decreasing as the distance of the cover to the center of the cube increases. Therefore, it suffices to study the two extreme cases, as we have in the paper.

---

> ### Author Response · Authors · 2025-11-21
>
> > Efficiency and practicality of ChunkySeal.ChunkySeal reaches higher bit capacity but at the cost of ~760M parameters. How feasible is this in deployment scenarios? Could lighter architectures (e.g., tiled embeddings, structured transforms) achieve similar performance?
>
> We would like to clarify that we had a mistake in the manuscript as per the number of parameters. 760M refers to a smaller 512bit model. The actual number for the 1024 bit model is roughly 1.8B parameters. As mentioned in our limitations, a model of this size cannot be deployed directly at scale. While this correction is somewhat tangential to the main scientific discussion, we wanted to ensure full transparency.
>
> The goal of us training ChunkySeal was to demonstrate that a capacity of 1024 bits with such quality and robustness constraints is at all feasible (which was not obvious). Achieving this capacity in a parameter-efficient manner is out-of-scope for this work. That being said, there are several things one can do, including the architectural changes you proposed, or distilling the model into a smaller one.
>
>
> > Lack of in-depth analysis on model failure causes : While VideoSeal’s underperformance is empirically demonstrated, the architectural or optimization bottlenecks (e.g., skip connections, normalization, bandwidth limits) are not deeply dissected.
>
> > Failure analysis of VideoSeal : What specifically prevents VideoSeal from scaling beyond 1024 bits—optimization instability, insufficient representation capacity, or architectural bottlenecks? Any diagnostic results (e.g., layer-wise activation spectra) to support this?
>
> > Theoretical limits vs. hyperparameter tuning : If a theoretical limit exists, why can’t it be reached through simple hyperparameter sweeps (e.g., reconstruction loss weight )? Is the gap due to optimization dynamics or representational constraints? A quantitative analysis (e.g., singular value decomposition of the embedding mapping) would clarify this.
>
> In-depth analysis of the possible architectural failings of the VideoSeal architecture (applicable to most other similar open-source models) would be a massive undertaking in of itself and is out-of-scope for the present paper. Still, we showed that a linear architecture easily learns it. That points to it not being a problem with the choice of optimizer hyperparameters. It is very likely that models that are even larger would be able to achieve more than 1024 bits with the same architecture, though this scaling quickly becomes unfeasible. The need to look for more parameter-efficient architectures is precisely the call to the research community we hope to make with this work.

---

### Official Review · Reviewer_Rj44 · 2025-10-25

**Soundness:** 3
**Presentation:** 4
**Contribution:** 3
**Rating:** 6
**Confidence:** 4

**Summary:**

This paper introduces a formalization of watermarking capacity and shows how many DL approaches to watermarking do not achieve this capacity. This work then proposes a new methodology that is able to use 1024 bits encoded.

**Strengths:**

This paper derives a first principal approach to understanding a fundamental question in watermarking: the theoretical limits of capacity that images can hold/embed under image quality and also robustness. Current literature usually uses on the order of 100-200 bits which is sufficient for many cases but this work highlights that this is under-represented. The authors show that using up to 1024 bits in practice has little to no performance drop.

**Weaknesses:**

I think that the current suite of attacks are kind of basic. I would ideally like to see some more modern attacks (regeneration, rinsing, and maybe even a combination of a lot of attacks). I think that these settings can really test the robustness of the method.

**Questions:**

- I would be curious to understand the theoretical formulation of a combination of attacks.
- I would also like to see if there is a principled way to understand regeneration attacks in your current framework.
- (The regeneration/other tests I asked for I mostly care about for empirical validation/comprehensiveness.)

---

> ### Author Response · Authors · 2025-11-21
>
> Thank you for your review!
>
> As far as more modern attacks are concerned, theoretical analysis would likely be unfeasible. As you have probably seen in the paper, the linear case already requires quite extensive and careful theoretical treatment because of the non-linearities involved with the quantization and pixel value clipping. Any more advanced distortions would require more simplification and approximation, and hence will likely offer little further insight. Regeneration attacks, where we would need to analyse the regeneration model (which is typically quite a large neural network), would be especially futile. As far as combination of attacks is concerned, we could not find an elegant bound that provides any usable insight but are open to the possibility one might exist.
>
> This being said, we conducted additional robustness evaluations of ChunkySeal and VideoSeal with a combination of attacks, more advanced attacks and attacks, as you can see in the tables below.
>
> | Attack                                | ChunkySeal BitAcc (1024 bits) | VideoSeal BitAcc (256 bits) |
> |---------------------------------------|-----------------------------|---------------------------|
> | Combo (JPEG 40, Crop 71%, Bright 0.5) | 57.6%                       | 70.6%                     |
> | Combo (JPEG 60, Crop 71%, Bright 0.5) | 66.2%                       | 74.0%                     |
> | Combo (JPEG 80, Crop 71%, Bright 0.5) | 82.0%                       | 77.6%                     |
> | Emoji overlay (size 32px)             | 100.0%                      | 99.6%                     |
> | Emoji overlay (size 48px)             | 100.0%                      | 99.6%                     |
> | Emoji overlay (size 64px)             | 100.0%                      | 99.6%                     |
> | Emoji overlay (size 96px)             | 100.0%                      | 99.6%                     |
> | Meme text (padding 0.08)              | 74.0%                       | 89.0%                     |
> | Meme text (padding 0.12)              | 69.2%                       | 82.0%                     |
> | Meme text (padding 0.16)              | 64.2%                       | 76.2%                     |
> | Meme text (padding 0.20)              | 62.6%                       | 71.8%                     |
> | Text overlay (font size 18)           | 100.0%                      | 99.6%                     |
> | Text overlay (font size 24)           | 100.0%                      | 99.6%                     |
> | Text overlay (font size 36)           | 100.0%                      | 99.6%                     |
> | Text overlay (font size 48)           | 100.0%                      | 99.6%                     |

---

> > ### Comment · Reviewer_Rj44 · 2025-11-27
> > **Response**
> >
> > I appreciate the author's experiments and response. I agree that theoretical analysis seems quite difficult here for an extended set of attacks so I will not penalize. I will leave my score as is and retain a favorable review on this work.

---

### Official Review · Reviewer_GG8c · 2025-10-27

**Soundness:** 3
**Presentation:** 3
**Contribution:** 3
**Rating:** 6
**Confidence:** 4

**Summary:**

This paper investigates whether modern deep watermarking is approaching the capacity–quality–robustness limit. Rather than working within the classical Gel’fand-Pinsker problem [1], the authors adopt a geometric, high-dimensional grid view to analyze the trade-off between PSNR and capacity under perfect decoding, first in the noise-free case and then under linear distortions. Empirically, a handcrafted construction comes close to the PSNR-only bound, and an expanded model (Chunky Seal) achieves ~4× capacity while maintaining PSNR and robustness comparable to VideoSeal. These results suggest that current deep watermarking systems remain far from the achievable limit in terms of capacity at a given quality/robustness level.

[1] S. Gelfand and M. Pinsker, “Coding for channel with random parameters,” Prob. of Control and Inf. Th., vol. 9, no. 1, pp. 19–31, 1980.

**Strengths:**

1.	The paper tackles a fundamental and important question: after roughly several years of progress in deep learning–based watermarking, are we actually approaching the limit of the quality–robustness–capacity trade-off? Rather than starting from the classical information-theoretic setting, the authors proceed from a high-dimensional grid perspective and derive, step by step, the maximum information capacity, the capacity under a PSNR constraint, and the capacity under linear distortions.

2.	Empirically, a handcrafted watermark in the noise-free setting approaches the theoretical upper bound, while under noise the Chunky Seal model achieves higher capacity yet similar PSNR and robustness to Video Seal, further indicating current limits of deep learning watermarking performance.

3.	The theoretical development is reasonable and clear: it analyzes the limitations of deep models and articulates a plausible theoretical upper limit.

**Weaknesses:**

1.	The related work is not sufficiently comprehensive. The paper does not adequately cite and explain existing traditional information-theoretic analyses, making it hard to evaluate the advantages of the proposed geometric high-dimensional grid approach over prior, thoroughly studied capacity analyses from the information-theoretic perspective.

2.	The current capacity analysis remains limited to linear distortions; discussion of non-linear distortions is still quite limited.

3.	The observed capacity gains via tiling are unusual, yet the paper provides little analysis of why this phenomenon occurs.

4.	In the noise-free case, the paper proposes a handcrafted method that nearly attains the theoretical optimum; however, it remains unclear how to approach the theoretical capacity under noise.

5.	In the high-capacity experiments, the paper does not compare against LISO [1], which achieves 4 bpp at ~25 dB PSNR with near-100% accuracy in the noise-free case. A study of high-capacity watermarking should analyze and compare with this method.

[1] Chen X, Kishore V, Weinberger K Q. Learning iterative neural optimizers for image steganography. ICLR 2022.

**Questions:**

1.	For non-linear distortions, if a theoretical analysis is not feasible, do the authors have empirical methods to predict or measure the upper-bound capacity?

2.	The finding that tiling increases capacity is quite unexpected. Do the authors have an analysis of why this occurs? Why does direct training at high capacity tend to fail?

3.	The straightforward expansion of Video Seal yields a model roughly 11× larger, yet the capacity is still only 0.0052 bpp. This does not appear to be a viable path toward the paper’s proposed theoretical upper limits. What new design ideas do the authors have for future models?

---

> ### Author Response · Authors · 2025-11-21
>
> Thank you for appreciating that the question we study in this paper is fundamental and important and that our empirical results are in support of this goal, as well as for finding our theoretical results reasonable and clear. We will now answer your outstanding questions.
>
> > The related work is not sufficiently comprehensive. The paper does not adequately cite and explain existing traditional information-theoretic analyses, making it hard to evaluate the advantages of the proposed geometric high-dimensional grid approach over prior, thoroughly studied capacity analyses from the information-theoretic perspective.
>
> Thank you for the comment. We agree that the discussion of classical information-theoretic analyses needs to be strengthened. We adjusted the paper to include:
> * **An expanded literature review on Information-theoretic approaches** by providing a clearer account of the classical information-theoretic lineage from the Gel’fand–Pinsker framework, through Costa-style Gaussian embedding, to later extensions by Moulin, O’Sullivan, Merhav, and others. The revised section now outlines the assumptions these works rely on (additive or memoryless noise models, Euclidean distortion metrics, continuous signal spaces) and explains why such assumptions prevent these analyses from capturing robustness to many of the image distortions in practice. This addition clarifies how prior capacity results are derived and the limitations of their applicability.
> * **We added a dedicated subsection** in the extended literature review in the appendix that highlights the need for the geometric framework. In particular, we explain that classical analyses do not account for robustness to geometric distortions—cropping, resampling, rotations, affine and nonrigid warps—while our geometric framework is designed to handle these transformations and to provide capacity predictions that remain meaningful under more realistic attack models.
>
> We hope these additions clarify the relationship to prior work and make the contribution of our geometric approach easier to evaluate.
>
> > The current capacity analysis remains limited to linear distortions; discussion of non-linear distortions is still quite limited.
>
> > For non-linear distortions, if a theoretical analysis is not feasible, do the authors have empirical methods to predict or measure the upper-bound capacity?
>
> We considered the linear case sufficient as it covers all frequently used robustness constraints. As you can see, the linear case already requires quite extensive and careful theoretical treatment because of the non-linearities involved with the quantization and pixel value clipping. Studying non-linear distortions would require more simplification and approximation, and hence will likely offer little further insight. **While we considered empirical methods, they tended to require a combinatorial amount of samples to achieve any meaningful statistical significance and hence would not be feasible for any but the very low (<10) dimensions.**
>
> >The observed capacity gains via tiling are unusual, yet the paper provides little analysis of why this phenomenon occurs. The finding that tiling increases capacity is quite unexpected. Do the authors have an analysis of why this occurs? Why does direct training at high capacity tend to fail?
>
> Thank you for raising this question! **We have now expanded the Results section to provide a clear explanation of why tiling increases capacity in this particular setup**. The key is that tiling only works in the restricted illustrative setup of the experiments in Section 3, namely, having no geometric robustness requirements. This setup allows us to split the image into tiles, watermark each tile independently with a different secret and finally stitch the tiles together. At decoding time, we can again split into tiles, feed each independently into the decoder and concatenate the decoded messages. In other words, watermarking and tiling commute in this setup. That is because we know exactly where each tile will be at detection time. If one were to consider geometric robustness requirements, e.g., cropping or rotations, tiling would no longer work as we would not know how to split the transformed image at detection time. The experiments in Section 3 aim to show that VideoSeal (and likely all other current deep learning-based architectures) fail at this very simple setup and the simple tiling baseline is a very clear evidence of their limitation. For practical deployments though, if one needs geometric robustness, the naive tiling strategy would not work. Nevertheless, if combined with geometric image synchronization (e.g., Jovanović et al. 2025, Fernandez et al. 2025) tiling could be used to boost capacity even under some level of geometric perturbations.
>
> Jovanović et al., Watermarking Autoregressive Image Generation, arXiv:2506.16349
>
> Fernandez et al., Geometric image synchronization with deep watermarking, arXiv:2509.15208

---

> > ### Author Response · Authors · 2025-11-21
> >
> > >In the noise-free case, the paper proposes a handcrafted method that nearly attains the theoretical optimum; however, it remains unclear how to approach the theoretical capacity under noise.
> >
> > We understand that by “noise” you mean “robustness to augmentations”. In this case yes, we do not have any handcrafted method that approaches the theoretical capacity under arbitrary (linear) robustness constraints. However, developing such a method analytically would not be feasible. After all, if it were, then we would not need to train neural network models to do it.
> >
> > > The straightforward expansion of Video Seal yields a model roughly 11× larger, yet the capacity is still only 0.0052 bpp. This does not appear to be a viable path toward the paper’s proposed theoretical upper limits. What new design ideas do the authors have for future models?
> >
> > This is an excellent question. What our paper shows empirically is that current models can reach higher capacities when scaled, but also that this approach quickly saturates and is clearly not the right long-term solution. Rather than proposing a specific architecture, we provide a set of **sanity-check principles in the discussion section** that any next-generation watermarking model must satisfy, e.g., capacity scaling linearly with image size, predictable PSNR-capacity tradeoffs, and degradation under augmentations that matches geometric reasoning. Architectures that maintain resolution-scaled feature fields, avoid fixed latent bottlenecks, and use geometrically equivariant decoders are natural candidates for meeting these criteria.

---

### Author Response · Authors · 2025-11-21
**Summary of Enhancements Made During the Rebuttal Phase**

We thank the reviewers for their thoughtful feedback and for highlighting the key strengths of our work particularly the “intuitive and insightful” geometric framework and the clear articulation of a plausible theoretical upper limit (Reviewers GG8C, 851W), as well as the strong empirical design and practical demonstration of higher capacities in ChunkySeal (Reviewer 851W). We especially want to thank the Area Chairs, and, in light of their increased workload, we want help by highlighting the contributions of our work, as well as the revisions made in direct response to the reviewers’ comments.

**Key contributions:**
1. We are the first to offer theoretical bounds on the message capacity of image watermarking under geometric perturbations and JPEG compression. While prior works have provided information-theoretical bounds (as also highlighted by Reviewer GG8c), they are necessarily restricted to Gaussian channels and Gaussian perturbations. Instead, we use a much more flexible geometric approach.
2. Our bounds show that capacities orders of magnitude larger than what we see in current models are feasible. We offer extensive experiments validating these results, something that the reviewers appreciated.
3. Finally, we put theory to practice by training ChunkySeal, a scale-up of VideoSeal, with 1024 bit capacity and matching image quality and robustness. To the best of our knowledge, this is the largest watermarking model ever trained, and the likely the best model based on an open-source architecture.
4. Overall, with this paper, we want to communicate an optimistic message to the deep watermarking community: there is plenty of space for ambitious and exciting developments and we can make orders of magnitude better watermarking models.

With extensive theoretical and empirical results, a new SOTA model, and a strong message to the community, we believe our paper to be a perfect match for ICLR.

**Below is a concise summary of the revisions made in direct response to the reviews:**

1. **Expanded literature review (Reviewer GG8C):** We expanded the discussion of classical information-theoretic work, including Gel’fand-Pinsker, Costa, and Moulin-O’Sullivan, and added the missing steganography reference suggested by the reviewer. We clarified why these models (e.g., Gaussian or memoryless-noise assumptions) cannot capture modern robustness settings, and strengthened the appendix with additional motivation.
2. **Extra Robustness Experiments (Reviewer Rj44):** We included stronger combined robustness attacks such as emoji overlays, meme text, text overlays, and combined augmentations (e.g., JPEG + crop + brightness). These new robustness results are now fully incorporated into the paper and the rebuttal.
3. **Clearer discussion on scaling (Reviewers GG8C, 851W):** We clarified that scaling of ChunkySeal was mainly to probe feasibility (i.e., showing 1024 bits is possible), not as a final production-ready model in of itself, we also suggested ways to productionize such a large model, e.g., via distillation. In the revised paper, we placed this clarification beside our sanity checks for Watermarking Research, which state that principled watermarking models should (1) scale capacity with image size, (2) decrease capacity linearly with higher PSNR, (3) outperform linear or handcrafted baselines, and (4) show predictable drops under stronger augmentations

Should you have any outstanding questions, we would be happy to address them.

The authors

---

### Meta-Review · Area_Chair_rxUy · 2026-01-12

**Summary:**

This paper investigates the theoretical and practical limits of image watermarking capacity, demonstrating that current deep learning-based methods are far from reaching the upper bounds of what is theoretically possible. The authors present a geometric framework to derive capacity limits under PSNR and linear robustness constraints, and they empirically validate these findings by scaling an existing model (VideoSeal) to achieve 1024-bit capacity—four times higher than current standards—while maintaining comparable image quality and robustness.

Reviewers raise the following concerns:

Reviewer GG8c noted that (1)  the capacity analysis is limited to linear distortions, with little discussion of non-linear cases. (2) Questioned the unexplained capacity gains from tiling and the lack of a clear path to approach theoretical capacity under noise. (3) Pointed out the absence of a comparison with the high-capacity method LISO. (4) The related work section was insufficiently comprehensive, particularly regarding classical information-theoretic analyses, making it difficult to evaluate the advantages of the proposed geometric approach.

Reviewer Rj44 noted that: (1)  the suite of robustness attacks is basic and suggested including more modern or combined attacks (e.g., regeneration, rinsing) for a more comprehensive empirical validation. (2) Requested a theoretical formulation for combined attacks and their integration into the existing framework.

Reviewer 851W (1) critiqued the reliance on PSNR as a perceptual metric, suggesting that LPIPS or MS-SSIM would better reflect real-world constraints. (2) highlighted the limited formal robustness analysis, noting that some bounds were heuristic or overly conservative, with non-linear effects like quantization not fully addressed. (3) questioned the practicality of ChunkySeal due to its large parameter count (1.8B) and raised concerns about deployment feasibility. (4) requested deeper failure analysis of VideoSeal to understand why it cannot scale beyond 1024 bits and suggested diagnostic evaluations (e.g., singular value decomposition). (5) asked about the completeness of the geometric cases in the theoretical analysis and how the framework generalizes to real image formats like JPEG or PNG.

According to the comments, although this work contributes a meaningful idea, it fails to adequately address fundamental theoretical limitations and practical concerns raised by reviewers—including the reliance on lack of robust analysis under non-linear distortions, and a lack of comparison with SOTA methods, etc.

**Reviewer Concerns:**

For Reviewer GG8c's concerns, the authors addressed concerns about related work and tiling by expanding the literature review and clarifying the geometric assumptions. However, outstanding issues remain: no comparison was made with the high-capacity method LISO, no analytical approach was given for reaching capacity under noise or non-linear distortions, and the treatment of non-linear robustness remains limited to heuristic bounds.

For Reviewer Rj44's concerns, the authors satisfactorily addressed the request for more robustness experiments by adding results for combined and modern attacks (emoji overlays, meme text, etc.). However, they did not provide a theoretical formulation for combined attacks, leaving the theoretical framework still limited to linear or simple composite cases. The reviewer agreed that this is challenge.

For Reviewer 851W, the authors clarified points about parameter counts, image format generalization, and cover image cases. Still outstanding are the reliance on PSNR instead of perceptual metrics, the lack of in-depth failure analysis for VideoSeal, no diagnostic exploration of architectural bottlenecks, and no practical solution for the high parameter cost of ChunkySeal in deployment.

**Reviewer Scores:**

For reviewer GG8c: Although the authors’ substantial expansion of the related work and clear explanation of tiling’s limitations, the lack of comparison to LISO and the absence of a theoretical path to capacity under non-linear distortions likely would have prevented a major score increase, keeping the review cautiously positive or reducing the scores.

For Reviewer Rj44, having seen the authors’ thorough empirical response with additional robustness tests—including combined and modern attacks—Reviewer Rj44 would likely have maintained or slightly increased their score, moving from “marginally above acceptance” to a clearer acceptance recommendation. The reviewer explicitly noted satisfaction with the experimental additions and acknowledged theoretical difficulties, suggesting the rebuttal adequately addressed their primary concerns.

For Reviewer 851W: Reviewer 851W might have remained around “marginally below acceptance” even after discussion, as several core concerns—reliance on PSNR, lack of failure analysis, and high parameter costs—were not fully resolved. While some clarifications were provided, the outstanding theoretical and practical limitations would likely have left the reviewer unconvinced that the paper yet meets the bar for strong acceptance, though the constructive rebuttal may have softened their critique slightly.

---

### Decision · Program_Chairs · 2026-01-26

Reject